



# Plio-Pleistocene Perth Basin water temperatures and Leeuwin Current dynamics (Indian Ocean) derived from oxygen and clumped isotope paleothermometry

David De Vleeschouwer[1], Marion Peral[2], Marta Marchegiano[2], Angelina Füllberg[1], Niklas Meinicke[1],
5   Heiko Pälike[1], Gerald Auer[3], Benjamin Petrick[4], Christophe Snoeck[2,5], Steven Goderis[2], Philippe Claeys[2]

[1]MARUM - Center for Marine Environmental Sciences, University of Bremen, Klagenfurterstr. 2-3, 28359 Bremen, Germany
[2]Analytical, Environmental, and Geo-Chemistry, Vrije Universiteit Brussel, Pleinlaan 2, 1050 Brussels, Belgium
[3]Institute of Earth Sciences (Geology and Paleontology), University of Graz, Heinrichstraße 26, 8010 Graz, Austria
[4]Paleontology and Historical Geology, Kiel University, Ludewig-Meyn-Str. 14 R.12, 24118 Kiel, Germany
10   [5]Maritime Cultures Research Institute, Vrije Universiteit Brussel, Pleinlaan 2, 1050 Brussels, Belgium

*Correspondence to*: David De Vleeschouwer (dadevlee@gmail.com)





**Abstract.** The Pliocene sedimentary record provides a window into Earth's climate dynamics under warmer-than-present
boundary conditions. However, the Pliocene cannot be considered a stable warm climate that constitutes a solid baseline for
middle-road future climate projections. Indeed, the increasing availability of time-continuous sedimentary archives (e.g.,
marine sediment cores) reveals complex temporal and spatial patterns of Pliocene ocean and climate variability on astronomical
timescales. The Perth Basin is particularly interesting in that respect because it remains unclear if and how the Leeuwin Current
sustained the comparably wet Pliocene climate in West-Australia, as well as how it influenced Southern Hemisphere
paleoclimate variability. To constrain Leeuwin Current dynamics in time and space, this project constructed a new orbitally-
resolved planktonic foraminifera (*Trilobatus sacculifer*) stable isotope record ($\delta^{18}$O and clumped isotopes $\Delta_{47}$) for the Plio-
Pleistocene (4 – 2 Ma) interval of International Ocean Discovery Program (IODP) Site U1459. It complements an existing
TEX$_{86}$ record from the same site and similar planktonic isotope records from the Northern Carnarvon Basin (ODP Site 763
and IODP Site U1463). The comparison of TEX$_{86}$ and $\Delta_{47}$ paleothermometers reveals that TEX$_{86}$ likely reflects sea surface
temperatures (SST, 23.8 – 28.9°C), whereas *T. sacculifer* $\Delta_{47}$ calcification temperatures probably echo the state of the lower
mixed layer and upper thermocline at the studied Site U1459 (18.2 – 20.8°C). The isotopic $\delta^{18}$O gradient along a 19°S – 29°S
latitudinal transect, between 3.9 – 2.2 Ma, displays large variability, ranging between 0.5 and 2.0‰, whereby a low latitudinal
gradient is indicative of a strong Leeuwin Current and vice versa. These results challenge the interpretation that suggested a
tectonic event in the Indonesian Throughflow as the cause for the rapid steepening of the isotopic gradient (0.9 to 1.5‰) around
3.7 Ma. The tectonic interpretation appears obsolete as it is now clear that the 3.7 Ma steepening of the isotopic gradient is
intermittent, with flat latitudinal gradients (~0.5‰) restored in the latest Pliocene (2.9 – 2.6 Ma). Still, the new analysis affirms
that a combination of astronomical forcing of wind patterns and eustatic sea level controlled Leeuwin Current intensity. A
period of relatively weak Leeuwin Current between 3.7 and 3.1 Ma is advocated; a time interval also marked by cooler
conditions throughout the Southern Hemisphere. In conclusion, the intensity of the Leeuwin Current and the latitudinal position
of the subtropical front are rooted in the same forcing: Heat transport through the Indonesian Throughflow (ITF) valve
propagated to the temperate zone through Indian Ocean poleward heat transport. The common ITF forcing explains the
observed coherence of Southern Hemisphere ocean and climate records.



# 1 Introduction

Despite climate aridity in large portions of the continent, Australia is habitable because of the many boundary currents
surrounding the continent. The Leeuwin Current is one of those boundary currents, flowing southward along the Northwest
Shelf of Australia (NWS), across the Carnarvon and Perth Basins, ultimately rounding Cape Leeuwin to flow further east into
the Great Australian Bight. Leeuwin Current intensity varies throughout the year, with strongest flow in austral winter when
the latitudinal steric height gradient is steepest (Fig. 1) (Godfrey and Ridgway, 1985; Church et al., 1989; Pearce, 1991; Waite
et al., 2007; Cresswell and Peterson, 2009; Ridgway and Godfrey, 2015). Thereby, the Leeuwin Current acts as a moisture and
heat source for the Mediterranean-like climate around Perth, characterized by wet and mild winters. Throughout the Neogene,
rainfall patterns in Western Australia experienced severe regime shifts on million-year timescales. Groeneveld et al. (2017)
describe an extremely arid middle Miocene based on sabkha-like sediments found on the Northwest Shelf of Australia. Tagliaro
et al. (2018) corroborate this climate interpretation by proposing sea level fall and regional aridity as causes for Miocene karst
in the Northern Carnarvon Basin. Throughout the late Miocene, a northward shift of the Westerlies allowed for a progressive
increase in precipitation in southwest Australia (Groeneveld et al., 2017). At the same time, the siliciclastic Bare Formation
was deposited on the NWS in a phase of deltaic margin progradation, thus correlating with increasing humidity in the northwest
Australian hinterland (Tagliaro et al., 2018). By the latest Miocene and throughout most of the Pliocene, Western Australia
was entirely governed by a wet or seasonally wet climate, i.e., the so-called "Humid Interval" (Christensen et al., 2017;
Karatsolis et al., 2020). The switch from an arid Miocene to a wet Pliocene is also reflected in fossil pollen records from
southern Australia (Sniderman et al., 2016). Finally, in the late Pliocene and early Pleistocene, climate transitioned back
towards a more arid and more seasonal precipitation regime, albeit with important glacial-interglacial variability and more
humid conditions during interglacials (Dodson and Ramrath, 2001; Dodson and Macphail, 2004; Fujioka et al., 2009; Gallagher
et al., 2014; Stuut et al., 2014; Kuhnt et al., 2015; Stuut et al., 2019; He and Wang, 2021). The Late Pliocene aridification of
Australia promoted C4 over C3 photosynthesis and probably constitutes the underlying reason for the late C4 expansion in
Australia, compared to other continents (Andrae et al., 2018).

While the million-year-scale Neogene hydroclimate evolution of Western Australia is relatively well constrained and
substantiated by multiple lines of evidence, important open questions remain regarding the temporal and causal relationships
between Australian hydroclimate evolution and Leeuwin Current dynamics. Cane and Molnar (2001) hypothesize that east
African aridification at 3 - 4 Ma was caused by ITF restriction, assuming a switch in the source of surface flow through the
Indonesian seaway from South to Nord Pacific waters. This publication triggered an increased research interest in the causal
relationship between Indo-Pacific paleoceanography and paleoclimate change. Karas et al. (2009) supports this hypothesis,
but points out that the switch likely took place at the subsurface level rather than at the surface. Using a general circulation
model, Krebs et al. (2011) links the end of the "Humid Interval" and the observed Late Pliocene desertification of Australia to
a reduction in ITF transport. In the same year, Karas et al. (2011) were the first to explicitly link the Late Pliocene aridification



of Western Australia to Leeuwin Current weakening in response to a reduced ITF. In 2015, IODP Expedition 356 "Indonesian Throughflow" cored seven sites along Australia's western margin and obtained long, time-continuous sediment sequences that have been used to chart Pliocene-to-recent Leeuwin Current dynamics on astronomical timescales and to link those dynamics to the climate evolution of western Australia. Auer et al. (2019) and De Vleeschouwer et al. (2018, 2019) focus on Late

Pliocene glacial-interglacial variability around Marine Isotope Stage M2 (MIS M2, 3.3 Ma) and suggest that Pliocene aridity is punctuated during glacials when the Leeuwin Current was relatively weak. Contrary, during strong interglacials, winter aridity in northwest Australia is alleviated by a more active Leeuwin Current, which acts as a moisture source. Organic geochemistry TEX$_{86}$ studies reveal three important SST cooling steps at sites under the influence of the Leeuwin Current at 3.3 – 3.1 Ma, 1.7 – 1.5 Ma, and 0.65 Ma (Petrick et al., 2019; Smith et al., 2020). These authors interpreted the cooling steps

to result from a progressive weakening of the Leeuwin Current in response to ITF constriction, at least during glacial periods. This interpretation neatly accords with the gradual aridification of western Australia during this period. Nevertheless, He et al. (2021) challenge this paradigm: They use zonal and meridional inter-site temperature gradients and productivity reconstructions to postulate that the Leeuwin Current became stronger -not weaker- after the Mid-Pleistocene Transition. More work is needed to scrutinize the He et al. (2021) interpretation: by clarifying the increasing discrepancies between $U^{K'}_{37}$ and

TEX$_{86}$ at their studied Site U1461 over the last 1 Myr, and by explaining the lack of correlation between their records and the benthic isotope stack.

On glacial-interglacial timescales, the Pleistocene behaviour of the Leeuwin Current is comparable to that of the Pliocene, as it oscillates between two contrasting states. Interglacials were characterized by a stronger Leeuwin Current as higher eustatic

sea levels allow for better connected shallow ITF pathways, for example sourcing the Leeuwin Current over a flooded Sahul Shelf in northern Australia. Contrary, glacials were marked by shelf exposure along the coast of Western Australia, reduced ITF volume, and a stronger West Australian Current flowing opposite to the Leeuwin Current, all contributing to a weakening of the Leeuwin Current (Wyrwoll et al., 2009; Spooner et al., 2011; Petrick et al., 2019). Auer et al. (2021) refine this picture: A glacial sea level drop exceeding -45 m would expose the Sahul shelf and cut off the Leeuwin Current from an important

shallow-water source area. A weaker Leeuwin Current, in turn, would allow for the upwelling of sub-Antarctic mode waters onto the Australian shelf and ultimately enhanced organic carbon burial. The coupling between Leeuwin Current intensity and glacial eustasy is also reflected in planktonic $\delta^{18}$O gradients along the Leeuwin Current pathway (19°S – 29°S): A steep gradient is observed when Leeuwin Current is weak during glacials, and vice versa during interglacials (De Vleeschouwer et al., 2019). However, the modelling results displayed in Figure 2 suggest quite the opposite. When the Pliocene glacial

simulation "*Large M2*" (Fig. 2b; Dolan et al., 2015) is compared to the warm-orbit Pliocene simulation "*PlioMax*" (Fig. 2a; Prescott et al., 2018), glacial cooling in the Carnarvon Basin seems to be more severe than in the Perth Basin (Fig. 2c). This pattern implies a flatter SST gradient during glacials compared to interglacials and is thus the exact opposite of what is observed in proxy data. The data-model discrepancy likely occurs because these models do not consider the effects of shelf exposure and a sea level-induced reduction in shallow-water ITF transport. Relative sea level consequently plays an important role in



feeding and directing the Leeuwin Current, but also suggests that Leeuwin Current intensity is sensitive to small changes in boundary conditions. This paper evaluates the driving factors of Leeuwin Current intensity on astronomical timescales by extending the planktonic $\delta^{18}O$ gradient approach across the Plio-Pleistocene boundary. Moreover, it compares $TEX_{86}$ and clumped isotope paleothermometers at IODP Site U1459 to constrain absolute temperatures (and eventually heat transport) throughout the Plio-Pleistocene. Both paleothermometers have their pitfalls: The origin of the $TEX_{86}$ signal is still debated,

while the $^{13}C^{18}O^{16}O$ isotopologue is so uncommon that clumped-isotope paleothermometry still deals with large uncertainties. Per contra, the uncertainty on clumped-isotope measurements is largely random, and therefore averaged measurements provide meaningful results, even when error bars are wide. This makes clumped-isotope paleotemperatures a useful benchmark for comparisons with proxies that potentially face systematic biases (e.g. $TEX_{86}$). By comparing different ocean temperature reconstructions, our ambition is to obtain a better understanding of the links between Leeuwin current dynamics and the climate

evolution of the Southern Hemisphere.



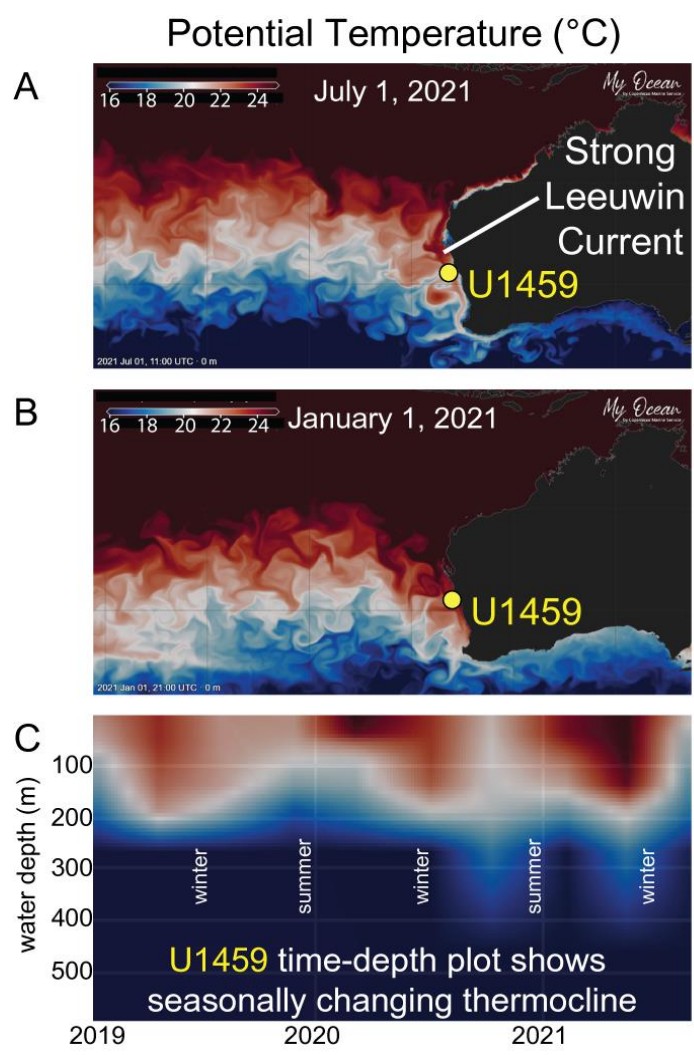

**Figure 1: Present-day oceanography of the eastern Indian Ocean.** The southward flowing Leeuwin Current is strongest in late autumn and winter. During that season, the Leeuwin Current generates large eddies, causing significant deepening of the mixed layer (down to ~200 m) and facilitating enhanced primary productivity (also see Suppl. Fig. B1). During austral summer, the Leeuwin Current is weaker, with a corresponding thinner mixed layer and reduced productivity. **(a)** Sea surface temperature on January 1$^{st}$, 2021 (austral winter), **(b)** Sea surface temperature on July 1$^{st}$, 2021 (austral summer). **(c)** Time-depth potential temperature plot for the Site U1459 locality. Sea surface temperatures vary seasonally between 20 and 24°C. Figures generated using E.U. Copernicus Marine Service Information with data from GLOBAL_ANALYSIS_FORECAST_PHY_001_024 (Zammit-Mangion and Wikle, 2020).



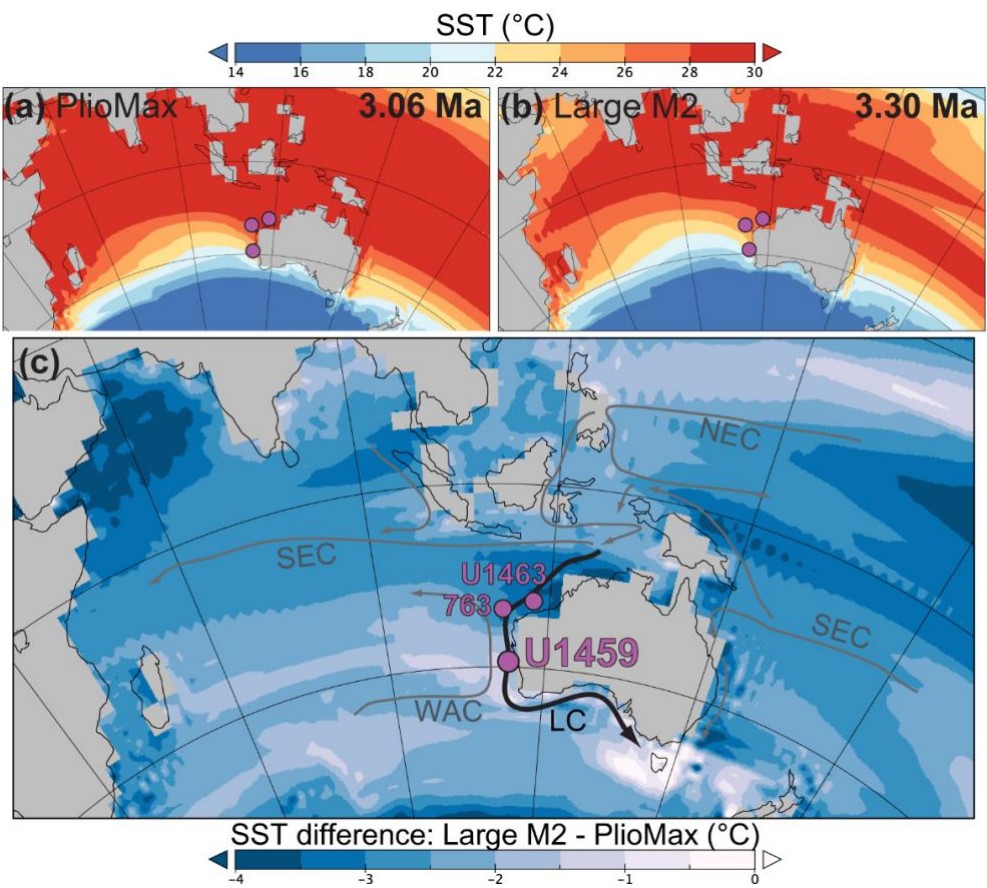

**Figure 2: Pliocene modelled sea surface temperatures (SST). (a)** Warm Pliocene simulation "*PlioMax*" for Marine Isotope Stage (MIS) K1 (3.06 Ma orbit & 405 ppm $CO_2$; Prescott et al., 2018). **(b)** Cold Pliocene simulation "*Large M2*" for MIS M2 (3.30 Ma orbit & 280 ppm $CO_2$; Dolan et al., 2015). **(c)** The SST difference between the cold and warm Pliocene simulations suggests that the latitudinal SST gradient along the Leeuwin Current (LC) pathway is reduced during glacials and steepens during interglacials. However, paleoceanographic proxy data from De Vleeschouwer et al. (2019) suggest the opposite: A steeper gradient during glacials compared to interglacials. NEC = North Equatorial Current. SEC = South Equatorial Current. WAC = West Australian Current.

| Calibration | A | B |
|---|---|---|
| Meinicke et al. (2020) | $40.7 \times 10^3$ | 0.1385 |
| Peral et al. (2018) | $37.0 \times 10^3$ | 0.181 |
| Anderson et al. (2021) | $39.1 \times 10^3$ | 0.154 |

**Table 1:** Clumped isotope paleothermometry calibration parameters for Eq. (1), using the I-CDES 90°C scale. The A and B parameters of the Meinicke et al. (2020) calibration were re-calculated from the CDES to the I-CDES 90°C framework by Meinicke et al. (2021). The Peral et al. (2018) calibration parameters were re-calculated by Marion Peral. A re-calculation of the Anderson et al. (2021) calibration was not necessary, as they have been originally defined in the I-CDES 90°C framework.





## 2 Materials and Methods

### 2.1 IODP Site U1459 (Perth Basin)

IODP Site U1459 (192 m water depth) lies in the northern part of the Perth Basin and is the southernmost site drilled in the IODP Expedition 356 latitudinal transect (Figs. 1, 2). The main objective of drilling Site U1459 was to obtain insight into the pre-Quaternary history of the Houtman Abrolhos reef complex and Leeuwin Current. In this work, the portion of Site U1459 between 55.65 – 106.41 meter composite depth (mcd) is studied, which corresponds to lithostratigraphic Units III and IV in Gallagher et al. (2017). Unit IV is an unlithified, cream to light brown packstone, ascribed to an outer shelf to upper bathyal area. Unit III is an alternatingly light gray – greenish-gray unlithified to partially lithified packstone to grainstone. Compared to Unit IV, Unit III is defined by a marked increase in glauconite content and bioclasts (benthic foraminifera, bivalves, echinoderms, bryozoans, and gastropods). Benthic foraminifera assemblages ascribe Unit III to a middle to outer shelf environment.

De Vleeschouwer et al. (2019) published several proxy series for Site U1459, including X-Ray Fluorescence (XRF)-derived element ratios, TEX$_{86}$ paleotemperatures, and planktonic $\delta^{18}$O and $\delta^{13}$C. The cyclostratigraphic age model in that paper consists of 15 age-depth tie points between 64 and 161 mcd, corresponding to 2.51 and 5.46 Ma. Here, we extend the studied interval into the early Pleistocene, which necessitates the addition of three age-depth tie points using a similar tuning approach as in De Vleeschouwer et al. (2019). As a result, the U1459 age-depth model now extends to 51.42 mcd, equivalent to 1.93 Ma (Suppl. Table A1, see also §3.1.).

### 2.2 IODP Site U1463 and ODP Site 763 (Northern Carnarvon Basin)

All new proxy records presented in this manuscript come from IODP Site U1459. In addition, we use previously published proxy records from IODP Site U1463 and Ocean Drilling Program (ODP) Site 763 to calculate planktonic $\delta^{18}$O and SST gradients along the Australian West Coast. IODP Site U1463 (145 m water depth) was cored in the Northern Carnarvon Basin (Beagle sub-basin) in 2015 and quickly became a regional reference site for reconstructing Miocene-to-recent climate, ocean, and basin dynamics (e.g. Christensen et al., 2017; De Vleeschouwer et al., 2018; Tagliaro et al., 2018; Auer et al., 2019; Gurnis et al., 2020; Karatsolis et al., 2020; Mccaffrey et al., 2020; Smith et al., 2020; Groeneveld et al., 2021). The Pliocene interval of U1463 consists of homogeneous fine-grained mudstones with subordinate wackestone and packstone intervals, with benthic foraminifera assemblages indicative of a middle to outer shelf environment. ODP Site 763 (1367 m water depth) was also cored in the Northern Carnarvon Basin (Exmouth Plateau) in 1988 and consists of light-gray to white foraminifer-nannofossil ooze (Haq et al., 1990). The age-depth model is from Karas et al. (2011), with a minor adjustment in the youngest part of the record (Suppl. Table A2).

Both sites are located within the trajectory of the Leeuwin Current, at 18.96°S and 20.58°S, respectively (~595 km apart). Because of their proximity and congruous oceanographic setting, we combine proxy series from both sites to construct a



"Northwest Shelf of Australia" end-member to compare the newly generated U1459 data with and to calculate paleoceanographic gradients along the Leeuwin Current pathway.

## 2.3. Oxygen and carbon stable isotope analyses

We present a new dataset of 238 stable ($\delta^{13}C$ and $\delta^{18}O$) isotope analyses, measured on calcite tests of the shallow dwelling

planktonic foraminifer *Trilobatus sacculifer*. The new dataset connects to the younger end of the previously published Pliocene isotope record in De Vleeschouwer et al. (2019; N = 143) and extends it into the Pleistocene. The new measurements come from the stratigraphic interval between 82.9 and 55.65 mcd, and sampling occurred at a median spatial resolution of 15 cm according to the shipboard composite depth (mcd) scale, which resulted from a color-reflectance-based correlation between Holes U1459A and U1459B (Gallagher et al., 2017). Specimens were picked from the 315–355-µm-size fraction to avoid size

effects in $\delta^{18}O$ values (Elderfield et al., 2002) and to be methodologically consistent with other planktonic isotopic records in the study area (Karas et al., 2011; De Vleeschouwer et al., 2018; 2019). All samples were measured using a Finnigan MAT 251 gas isotope ratio mass spectrometer connected to a Kiel III automated carbonate preparation device at the Center for Marine Environmental Sciences (MARUM). Isotopic data are reported in standard delta-notation versus V-PDB. We calibrated all measurements against the in-house standard (homogenized Solnhofen limestone powder), which in turn is calibrated against

the NBS-19 reference material. Over the measurement period, the standard deviations (1σ) of the in-house standard (N = 90) were 0.04‰ for $\delta^{13}C$ and 0.06‰ for $\delta^{18}O$.

## 2.4. Clumped isotope thermometry in planktonic foraminifera

Clumped isotope measurements were carried out using calcite tests of *T. sacculifer* in the 315–355-µm size fraction. Specimens from up to four adjacent samples were pooled to obtain sufficient material for clumped isotope paleothermometry (150 – 350

specimens, 2-5 mg). Potential contaminants were then removed using a modified version of the cleaning protocol for foraminiferal Mg/Ca analysis of Barker et al. (2003), without the $H_2O_2$ treatment steps (Grauel et al., 2013; Peral et al., 2018). The clumped isotope analysis of the cleaned carbonate powders took place at the Vrije Universiteit Brussel, Belgium (AMGC-VUB lab), using a Nu Instruments Perspective-IS stable isotope ratio mass spectrometer (SIRMS) in conjunction with a Nu-Carb carbonate sample preparation system. This setup also includes a fully automated adsorptive trap purification system to

remove contaminants. Between 500 and 600 µg of carbonate, power reacts for 10 minutes at 70°C after the automated injection of 120 µL of $H_3PO_4$. The $CO_2$ gas produced is expanded into a water trap for 5 minutes, held at –95°C. Subsequently, the $CO_2$ gas moves through a PoraPak™ Q packed adsorption trap, held at -34°C for 25 minutes, and trapped into a liquid nitrogen ($LN_2$) cold finger. After a yield measurement utilizing a pressure transducer, the sample is transferred into the sample cold finger, for 3 minutes, within the inlet system of the mass spectrometer. The cold finger on the sample side of the dual-inlet acts

as a constant depletion volume once the gas has been expanded. This is matched by an identical depletion volume on the reference side of the dual-inlet system. This allows for a constant depletion rate of both sample and reference gas during the





data acquisition. Additionally, the absorptive trap is cleaned/degassed by active vacuum pumping at 150°C between sample extractions for 25 minutes to remove any contaminants that may have been trapped during the gas transfer. Reference and sample gases are alternatingly measured on six Faraday collectors (m/z 44-49) and analysed in 3 "blocks" of 20 cycles each

with a counting time of 20 seconds. Measurements sum up to 20 minutes of integration time per replicate. A zero (no-gas) background measurement and automatic peak centring (on m/z 45) are performed at the start of each sample measurement. The reference gas beam is pressure balanced to the sample gas beam and depletes evenly through matched length capillaries; initial beam balance (m/z=44) is set to 80 nA, weakening to approximately 45 nA. The reference side of the dual-inlet was refilled with reference gas every 7 analyses. Total analysis time (including the reaction, PoraPak purification, and integration

time) is approximately 1h 20min per sample. The subsequent sample starts its preparation while the mass spectrometer is still analysing the previous sample. Possible contamination is monitored on ETH1-4 standards by scrutinising $\Delta_{48\ \text{raw}}$ and $\Delta_{49\ \text{raw}}$ values for extraordinary deviations from the mean (Meckler et al., 2014). ETH standards were measured following the recommendations of Kocken et al. (2019) with a sample to standard ratio of 1:1. Analyses and results are monitored in the lab using the Easotope software (John and Bowen, 2016). The raw $\Delta_{47}$ values of sample-derived $CO_2$ were converted to the I-

CDES 90°C scale, using the most recent values for the ETH-1, ETH-2, and ETH-3 carbonate reference materials, and using an acid fractionation factor of -0.022‰ (Bernasconi et al., 2021). The raw measured $\Delta_{47}$ values were processed using the IUPAC isotopic parameters (Brand et al., 2010; Daëron et al., 2016; Petersen et al., 2019) within the ClumpyCrunch software (Daëron, 2021).

The *T. sacculifer* clumped isotope measurements from Site U1459 were converted into calcification temperatures, using three different calibration schemes, which all follow the form of Eq. (1).

$$\Delta_{47} = \frac{A}{T^2} + B \qquad (1)$$

The most suitable calibration for the dataset presented here is that of Meinicke et al. (2020), as it is specifically designed for planktonic foraminifera, comprising 14 different species. Another relevant calibration for this study is the clumped isotope

paleothermometry calibration by Peral et al. (2018), based on 9 planktonic and 2 benthic species. In fact, the Meinicke et al. (2020) calibration includes the data-sets of earlier calibrations by Peral et al. (2018) and Piasecki et al. (2019). The most recent Anderson et al. (2021) calibration has the advantage of covering a wide range of temperatures (0.5 – 1100°C). Yet, it deviates somewhat from the two foraminifera-specific calibrations. This is probably due to the large influence of only a handful of data points on the warm end of the calibration and we argue that calibrations focusing on the natural range of ocean temperatures

are better suited for the paleoceanographic purposes of this study. For these reasons, we adopt the Meinicke et al. (2020) calibration throughout this paper, using the other two as referrals. Note that the Peral et al. (2018) and Meinicke et al. (2020) regression parameters have been updated to match the new I-CDES 90°C scale (Bernasconi et al., 2021). The updated version of the latter calibration can be found in Meinicke et al. (2021).





## 3 Results

### 3.1. High-resolution oxygen isotope record and construction of the age-depth model

The new planktonic $\delta^{18}$O depth-series (obtained from 55.65 – 82.90 mcd) smoothly connects to the previously-published dataset (83.4 – 106.41 mcd; De Vleeschouwer et al., 2019): The two datasets align in terms of absolute values at the transition point, and the variance and isotopic range also remain similar between the younger and older interval (Fig. 3a). The previously published dataset exhibits positive $\delta^{18}$O values up to 0.68‰ around 88 mcd, which were linked to the global MIS M2 glaciation by De Vleeschouwer et al. (2019). A gradual trend towards more negative values characterizes the record in the last few meters above this level. The new dataset starts at about the point where the gradual trend is concluded. The first ~17 m of the new dataset (82.9 – 65.5 mcd) are marked by relatively negative (between 0 and -0.5‰) and low-variance isotope values. No distinct positive excursions occur in this interval. At 65.5 mcd, a rapid shift to more positive $\delta^{18}$O values occurs and this shift is accompanied by increased variability. This stratigraphic interval lies outside the range of the age-depth model that was constructed in De Vleeschouwer et al. (2019). Hence, to convert depth to age, it was necessary to extend the available age-depth model by three additional age-depth tie-points. It was not possible to use the Ca/Fe depth-series for this purpose because it has large data gaps in the upper interval (Fig. 3b). Instead, we aligned the planktonic $\delta^{18}$O record directly with a southern-hemisphere eccentricity-tilt-precession (ETP) composite (Laskar et al., 2004) and the LR04 benthic stack (Lisiecki and Raymo, 2005). As we look at astronomical forcing signatures within the Site U1459 $\delta^{18}$O signal and in the latitudinal isotopic gradient along the western coast of Australia later in this work, we recognize that this age-depth modelling approach carries some risk for circular reasoning. However, no viable alternative technique allows the construction of a reliable age-depth model with $10^5$-year temporal resolution (i.e., an order of magnitude greater than the biostratigraphic time resolution in this interval). Nevertheless, we minimize the potential impact of circular reasoning by limiting the alignment of the isotope record to the target curves to only two tie-points (one at MIS 92 and one at MIS 87). The third tie-point is a biostratigraphic tie-point at 1.93 Ma and thus does not impose any circular reasoning (Fig. 3d).





**Figure 3: IODP Site U1459 depth-series and age-depth model. (a)** Planktonic δ^18O depth-series, produced by merging the measurement series in De Vleeschouwer et al. (2019, N =143) with the new data presented here (N = 238). The stratigraphic positions of marked glacial

260 intervals (MIS 96, 100, M2, Gi-2, and Gi-4) are indicated. **(b-c)** XRF-derived Ca/Fe and K/Al depth-series, interpreted in De Vleeschouwer et al. (2019) as proxies for detrital input and aeolian kaolinite flux, respectively. Both element ratios are plotted on a logarithmic scale. **(d)** The age-depth model by De Vleeschouwer et al. (2019) is extended into the early Pleistocene by three additional age-depth tie-points (transparent squares). Biostratigraphic markers are indicated by brown diamonds.





### 3.2. Clumped isotope-based calcification temperature reconstructions.

All clumped isotope measurements (standards and samples) carried out in the framework of this study are summarized in Table 2. The ETH-1, ETH-2, and ETH-3 standards define the ICDES-90°C framework and, therefore, correspond to the values fixed by the InterCarb initiative (Bernasconi et al., 2021). The ETH-4 standard is used for quality control, as its $\Delta_{47}$ composition is known, but it is not used to define the ICDES-90°C isotopic framework. The measured value for the ETH-4 standard matches the expected value within error margins (± 1SE; Table 2). Throughout the studied interval of IODP Site U1459, eight samples have been measured for clumped isotope paleothermometry (PB01 through PB08). Sample measurements resulted in $\Delta_{47}$ compositions that range between 0.6104 and 0.6370‰. The availability of sample material directly determined the number of repeated aliquot measurements carried out per sample. Sample availability remains the main limiting factor for clumped isotope paleothermometry, as ~500 µg of sample material is required per aliquot in the AMGC-VUB lab. The reported uncertainties (reported here as ± 1SE) scale to the number of repeated measurements (with smallest uncertainties for the PB04 result determined using 9 aliquots, and highest uncertainties for the PB05 result with only 2 aliquots), as well as to $\Delta_{47}$ repeatability within individual sessions ($\sigma = 0.314$‰ for the session with PB03 and $\sigma = 0.173$‰ for the session with all other samples).

The $\Delta_{47}$ clumped isotope compositions translate to Perth Basin calcification paleotemperatures between 12 and 21°C when using the Meinicke et al. (2020) or Peral et al. (2018) calibrations. The Anderson et al. (2021) calibration produces paleotemperatures that are ~1°C cooler (Table 3). Compared to the available $TEX_{86}$-based SST reconstruction for IODP Site U1459, the clumped isotope temperatures are about 5°C cooler. This result is independent of which $TEX_{86}$ surface calibration is applied: BAYSPAR (Tierney and Tingley, 2014), or $TEX_{86}^{H}$ (Kim et al., 2010) (Fig. 4a). The exact origin of the $TEX_{86}$ signal remains controversial though: It could be a surface, a subsurface, or a combined signal. In the ITF and Leeuwin Current study region, most authors used $TEX_{86}$ as a proxy for surface temperatures (De Vleeschouwer et al., 2019; Petrick et al., 2019; Smith et al., 2020). However, He et al. (2021) used it as a proxy for sub-surface temperatures and Meinicke et al. (2021) inferred a mixed surface-subsurface signal. In this paper, we start from the assumption that $TEX_{86}$ is a proxy for surface temperatures, but we add that the BAYSPAR subsurface calibration yields temperatures that are ~3.3°C cooler compared to the surface calibration, while the Ho and Laepple (2016) subsurface calibration yields cool temperatures (13.7–15.9°C) that correspond to modern-day lower thermocline temperatures. Most of the clumped-isotope derived calcification temperatures intersect with the 5%-percentile of the $TEX_{86}$ BAYSPAR surface calibration (Fig. 4a). Sample PB06 constitutes an exception to this general relationship between both proxies, as the clumped-isotope calcification temperature is much cooler compared to the corresponding $TEX_{86}$ surface paleotemperature. We note, however, that sample PB06 corresponds to the early Pleistocene glacial MIS 96, characterized by the most positive $\delta^{18}O$ values in the studied interval (Fig. 4b). The clumped isotope paleotemperatures are also compared to a foraminiferal $\delta^{18}O$-based SST reconstruction (Fig. 5b), using the Perth Basin seawater $\delta^{18}O$ reconstruction presented in Figure 5a as an assumption. The $\delta^{18}O_{sw}$ has been obtained by superimposing the



Plio-Pleistocene glacial contribution to $\delta^{18}O_{sw}$ (Rohling et al., 2014) to the present-day isotopic composition of Perth Basin seawater (+0.8‰ V-SMOW, dashed line on Fig. 5a). This approach results in a high-resolution SST reconstruction with

temperatures ranging between 15 and 22°C. The $\delta^{18}O$-based reconstruction more closely aligns with the clumped-isotope temperatures (except for PB06) than with the TEX$_{86}$ reconstruction (Figs. 4, 5).

| Identifier | N | $\delta^{13}C$ (‰ V-PDB) | $\delta^{18}O$ (‰ V-PDB) | $\Delta_{47}$ (I-CDES 90°C) (‰ ± 1SE) | Comment |
|---|---|---|---|---|---|
| *Standards used for ICDES-90°C scaling* | | | | | |
| ETH-1 | 35 | 2.00 | -2.16 | 0.2052 | |
| ETH-2 | 36 | -10.17 | -18.76 | 0.2085 | |
| ETH-3 | 54 | 1.72 | -1.72 | 0.6132 | |
| *Standard used for quality-control* | | | | | |
| ETH-4 | 39 | -10.15 | -18.74 | 0.4471 ± 0.0075 | *this study* |
| | 945 | | | 0.4511 ± 0.0011 | InterCarb 2021 |
| *Samples* | | | | | |
| PB01 | 6 | 1.10 | -0.56 | 0.6111 ± 0.0071 | |
| PB02 | 8 | 1.30 | -0.06 | 0.6096 ± 0.0061 | |
| PB03 | 3 | 1.03 | -0.03 | 0.6175 ± 0.0157 | |
| PB04 | 9 | 1.47 | 0.21 | 0.6141 ± 0.0058 | |
| PB05 | 2 | 1.12 | -0.42 | 0.6105 ± 0.0122 | |
| PB06 | 3 | 0.95 | 0.09 | 0.6365 ± 0.0100 | |
| PB07 | 6 | 1.07 | -0.22 | 0.6144 ± 0.0065 | |
| PB08 | 3 | 1.00 | -0.44 | 0.6179 ± 0.0100 | |

**Table 2: Clumped isotope results.** ETH-1, 2, and 3 define the ICDES-90°C framework. Hence their $\Delta_{47}$ are identical to the values reported by Bernasconi et al. (2021). The ETH-4 standard is used for monitoring measurement accuracy. The availability of sample material allowed for 2 to 9 repeated measurements (~500 µg each). The standard error on sample measurements is calculated by dividing the corresponding session repeatability (σ) by the square root out of the number of aliquots ($\sqrt{N}$), with σ = 0.0314 for PB03, and σ = 0.0173 for all other samples. The $\delta^{18}O$ and $\delta^{13}C$ sample values reported in

this table were measured simultaneously with $\Delta_{47}$ in the AMGC-VUB lab.





| | $\delta^{18}O$ (‰ V-PDB) | $\Delta_{47}$ (I-CDES 90°C) (‰ ± 1SE) | Meinicke et al. (2020) (°C ± 1SE) | Peral et al. (2018) (°C ± 1SE) | Anderson et al. (2021) (°C ± 1SE) | $\delta^{18}O_{sw}$ (‰ V-SMOW ± 1SE) |
|---|---|---|---|---|---|---|
| PB_01 | -0.56 | 0.6111 ± 0.0071 | 20.3 ± 2.2 | 20.2 ± 2.4 | 19.3 ± 2.3 | 0.17 ± 0.49 |
| PB_02 | -0.06 | 0.6096 ± 0.0061 | 20.8 ± 1.9 | 20.7 ± 2.1 | 19.8 ± 2.0 | 0.78 ± 0.43 |
| PB_03 | -0.03 | 0.6175 ± 0.0157 | 18.3 ± 4.9 | 18.0 ± 5.4 | 17.3 ± 5.0 | 0.27 ± 1.08 |
| PB_04 | 0.21 | 0.6141 ± 0.0058 | 19.4 ± 1.8 | 19.1 ± 2.0 | 18.4 ± 1.8 | 0.74 ± 0.40 |
| PB_05 | -0.42 | 0.6105 ± 0.0122 | 20.5 ± 3.9 | 20.4 ± 4.3 | 19.5 ± 4.0 | 0.35 ± 0.85 |
| PB_06 | 0.09 | 0.6365 ± 0.0100 | 12.7 ± 2.9 | 11.9 ± 3.2 | 11.5 ± 3.0 | -0.85 ± 0.63 |
| PB_07 | -0.22 | 0.6144 ± 0.0065 | 19.3 ± 2.0 | 19.0 ± 2.2 | 18.3 ± 2.1 | 0.28 ± 0.45 |
| PB_08 | -0.44 | 0.6179 ± 0.0100 | 18.2 ± 3.1 | 17.9 ± 3.4 | 17.2 ± 3.2 | -0.17 ± 0.67 |

**Table 3: Clumped isotope paleothermometry.** Clumped-isotopes reconstructed calcification temperatures according to three different calibrations. The Meinicke et al. (2020) calibration is adopted throughout this work. The $\delta^{18}O_{sw}$ of seawater was reconstructed by solving the Erez and Luz (1983) equation, using the clumped-isotope calcification temperature and the foraminiferal $\delta^{18}O$.




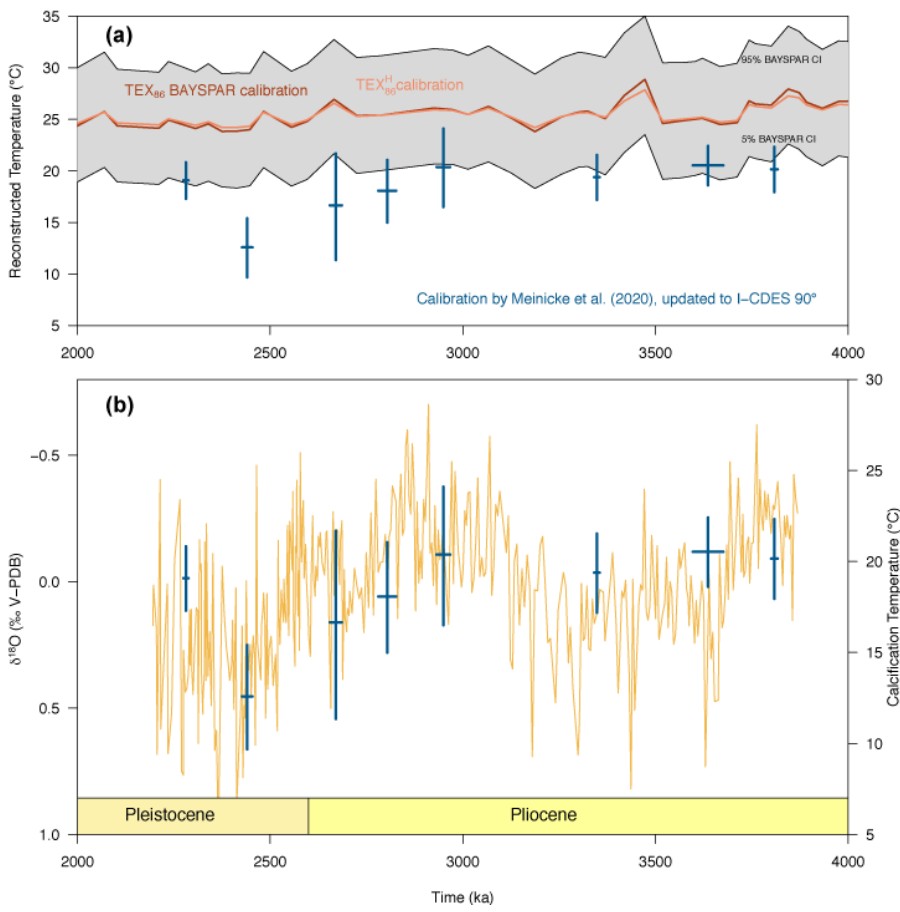

**Figure 4: (a)** Ocean temperature reconstructions based on TEX$_{86}$ (De Vleeschouwer et al., 2019) and clumped isotopes (this study) are offset by about 5°C. **(b)** Comparing the clumped isotope calcification temperature reconstruction to the U1459 planktic $\delta^{18}$O time-series reveals that the coolest $\Delta_{47}$ temperature corresponds to the heaviest $\delta^{18}$O values.




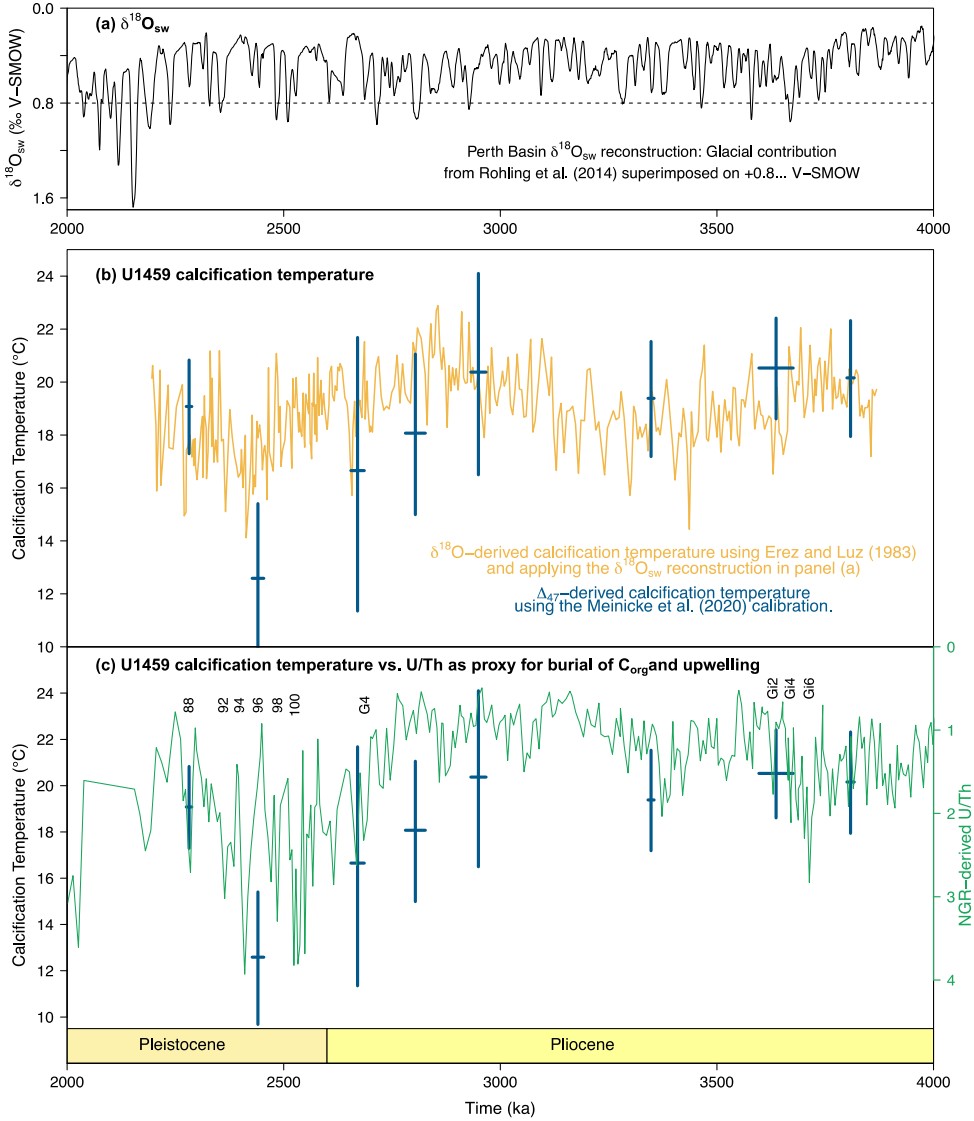

**Figure 5: (a)** Seawater $\delta^{18}O_{sw}$ reconstruction for the Perth Basin, constructed by superimposing the Rohling et al. (2014) glacial contribution to $\delta^{18}O_{sw}$ on the present-day isotopic composition (+0.8 ‰V-SMOW). **(b)** The U1459 planktic $\delta^{18}O$ time-series can be converted to a high-resolution calcification temperature time-series by using the Erez and Luz (1983) equation and adopting the $\delta^{18}O_{sw}$ reconstruction in panel (a). The high-resolution $\delta^{18}O$-derived temperature series is compared to the lower-resolution $\Delta_{47}$-derived paleotemperatures. **(c)** $\Delta_{47}$-derived paleotemperatures are compared to the U1459 core-based U/Th series (note the reversed y-axis). We use U/Th as a proxy for $C_{org}$ burial and upwelling of cold sub-Antarctic mode waters (SAMW) under a weak Leeuwin Current regime (proxy justification in Auer et al. (2021)). We note that the cold MIS 96 calcification temperatures might be in part explained by upwelling of lower thermocline waters.



## 4 Discussion

### 4.1. Discrepancy between TEX$_{86}$ sea surface and clumped-isotope calcification temperatures

The TEX$_{86}$ SST reconstruction at Site U1459 is slightly warmer than the simulated annual average sea surface temperatures

of the Pliocene climate models shown in Fig. 2a-b (20 - 24°C). At the same time, the TEX$_{86}$ surface temperatures are clearly and significantly warmer than the clumped-isotope calcification temperatures obtained by analyzing planktonic foraminifer *T. sacculifer*. The large temperature difference between these proxies demands a discussion of the processes that may underpin the discrepancy. The TEX$_{86}$ temperature reconstruction was assessed as reliable in De Vleeschouwer et al. (2019) because glycerol dialkyl glycerol tetraethers (GDGTs) were confirmed to be of marine origin, and all control parameters had values

within the recommended range for reliable paleo-SST reconstructions based on TEX$_{86}$. In this study, we adhere to this assessment. Indeed, the TEX$_{86}$ reconstruction for the Plio-Pleistocene with temperatures between 23.8 – 28.9°C seems reasonable compared to the present-day mean annual temperature of about 22.8°C (Reynolds et al., 2002). Nonetheless, we point out at least two potential warm-biases in the U1459 TEX$_{86}$ SST reconstruction. First, TEX$_{86}$ SST reconstructions along the Leeuwin Current pathway might suffer from a seasonal bias towards summer temperatures, although the effect of

seasonality on TEX$_{86}$ is poorly understood. A seasonal warm bias is supported by the observation that core-top TEX$_{86}$ SST are 3°C warmer than modern mean annual SST at Site U1463 in the Carnarvon Basin (Smith et al., 2020). A similar bias has been observed at Perth Basin Site U1460 by Benjamin Petrick (pers. comm.). The TEX$_{86}$ seasonal warm-bias along the Leewuin Current pathway could be linked to the oligotrophic conditions that characterize surface waters off the Australian west coast during summer (when Leeuwin Current is weak and the eddy-induced nutrient flux is minimum), allowing Thaumarchaeota

to dominate and thrive. A second potential TEX$_{86}$ warm-bias in the Perth Basin arises from the southward displacement of sinking particles, as Benthien and Müller (2000) proposed for the western Argentine Basin. Using a typical sinking rate of 100 m/day, a Leeuwin Current velocity of 50 cm/s, and a paleo-water depth of ~500 m, we estimate the order of magnitude of possible southward displacement of particles at several hundred kilometres. This potential warm-bias might not be negligible because of the rather steep latitudinal temperature gradient offshore Southwestern Australia but is insufficient to explain

temperature differences >1°C. Moreover, the *T. sacculifer* clumped-isotope paleothermometer would be subject to a similar, yet smaller, effect (Takahashi and Be, 1984). We conclude that there are no obvious reasons to doubt the reliability of the Plio-Pleistocene U1459 TEX$_{86}$ series as a tool to reconstruct Perth Basin SSTs. Minor warm-biases might have influenced the TEX$_{86}$ temperature reconstruction, but these biases are smaller than the observed temperature difference between the TEX$_{86}$ and clumped-isotope paleothermometers.


Similar to TEX$_{86}$, the clumped isotope paleothermometer is also subject to potential overprinting and biases. Indeed, we cannot exclude a diagenetic impact on $\Delta_{47}$. Yet, our confidence in the data is strengthened by the preservation of the selected *T. sacculifer* specimens (Suppl. Fig. C1), which was classified as moderate to good (Gallagher et al., 2017), the internal coherence of the $\delta^{18}O$ and $\Delta_{47}$ series (Fig. 4b, 5b), and the observation that the calculated $\delta^{18}O_{sw}$ values (Table 3) are within the expected



range for the Pliocene Perth Basin (Fig. 5a). The isotopic consonance between $\delta^{18}O$ and $\Delta_{47}$ at Site U1459 constitutes an important indicator, suggesting that it is unlikely that isotopic scrambling within the calcite of the analysed foraminiferal tests occurred. Instead, we explain the $TEX_{86}$-clumped isotope temperature difference by considering that the clumped-isotope calcification temperatures reflect water temperatures at the bottom of the mixed-layer rather than true surface temperatures. The mixed layer offshore southwest Australia can be fairly deep, down to 200 m during austral winter (Fig. 1c). The observed

discrepancy between proxies would thus reflect the temperature difference between the bottom of the mixed layer (clumped isotopes) and the top of the water column ($TEX_{86}$). This assessment is underpinned by the results presented by Zhang et al. (2019). In a study in the Indonesian Throughflow region, they found that *T. sacculifer* calcified at the bottom of the mixed layer or the upper thermocline. These authors suggest that the temperature difference between *T. sacculifer* and *Globigerinoides ruber* sensu strictu, a true surface-water dweller, could be used as a proxy for mixed-layer depth in the region.

Their results were largely in line with earlier work by Rippert et al. (2016), who derived a *T. sacculifer* calcification depth of ~120 m in the West Pacific Warm Pool, which is at the bottom of the mixed layer or the top of the thermocline. Nevertheless, the extension of the vertical habitat of *T. sacculifer* into the upper thermocline remains controversial, for example because it contradicts an earlier study off Indonesia: Mohtadi et al. (2011) calculated much shallower calcification depths (between 20 – 75 m) for *T. sacculifer* within the mixed layer. Regardless of the controversy, it becomes increasingly clear that *T. sacculifer*

has a wider vertical range than *G. ruber*. For example, a large-scale analysis of plankton net haul data from the subtropical eastern North Atlantic reports average living depths for *T. sacculifer* between 15 – 200 m (Rebotim et al., 2017), and a study in the Kuroshio Current system also found *T. sacculifer* down to 200 m water depth (Kuroyanagi and Kawahata, 2004). Overall, these recent planktonic foraminiferal habitat studies support the notion that *T. sacculifer* is a species able to adapt its temporal (lunar cycle, seasonal cycle) and spatial (water depth) habitat to local oceanographic conditions (Jonkers and Kučera, 2015,

2017; Kretschmer et al., 2018). The temperature discrepancy between $TEX_{86}$ and clumped isotopes could be largely explained by *T. sacculifer* descending in the water column during austral winter, when the mixed layer is up to 200 m thick (Fig. 1), whereas $TEX_{86}$ reflects surface temperatures with a potential summer-bias. In addition, isotopic proxies measured on *T. sacculifer* calcite tests could also be affected by a cold-bias through the addition of gametogenic calcite. This effect has been described by Bé (1980), who noticed that living *T. sacculifer* shells collected during towing were smaller than surface sediment

specimens. The size difference was ascribed by Duplessy et al. (1981) to the addition of calcite (gametogenic calcification) during their descent below the euphotic zone. This crust of secondary calcite is typically formed below the thermocline and can contribute up to ~20% of the post-gametogenic shell. Hence, gametogenic calcification can impose a significant cold-bias on the isotopically-reconstructed temperatures and result in unexpectedly deep apparent calcification temperatures (Wycech et al., 2018).


The above explanation is inadequate to account for the PB06 clumped-isotope temperature of 12.7 ± 2.9°C: Even if this measurement represents an upper thermocline temperature, it is too cold and outside of the temperature range that is tolerated by *T. sacculifer* (>14°C; Bijma et al., 1990). A *T. sacculifer* calcification temperature around 15°C, as suggested by the $\delta^{18}O$-




based temperature reconstruction at the same stratigraphic position (Fig. 5b), seems more plausible, albeit still at the lower end

of the tolerated temperature range. We note that sample PB06 corresponds to the early Pleistocene glacial MIS 96, which is

the third severe glacial interval after the intensification of Northern Hemisphere glaciation. A marked cooling, potentially

amplified by a sea-level-related positive feedback mechanism and the upwelling of sub-Antarctic mode waters (De

Vleeschouwer et al., 2019; Auer et al., 2021), is thus in line with expectations. Furthermore, the $\delta^{18}$O-based ~15°C temperature

is within the error margin on the clumped-isotope temperature. For all these reasons, we continue to interpret the PB06

clumped-isotope temperature to be unrealistically cold. While, at the same time, we maintain the interpretation that MIS 96

represents the coldest *T. sacculifer* calcification temperatures in the studied Plio-Pleistocene interval, characterized by a rapid

and marked cooling of the lower mixed layer and upper thermocline. With that perspective, it is interesting to note that glacial

stages MIS 100, 98, and 96 are marked by high U/Th (Fig. 5c). Here, we employ the U/Th ratio as a proxy for organic carbon

burial, following the proxy-interpretation by Auer et al. (2021) for the mid-Pleistocene interval of nearby Site U1460.

Enhanced $C_{org}$ burial during intense glacial stages is linked to the upwelling of nutrient-rich sub-Antarctic Mode waters onto

the western Australian shelf. The upwelling of those waters triggered enhanced productivity and caused a contraction of the

mixed layer and a marked steepening of the thermocline. We propose that upwelling of sub-Antarctic Mode waters at times of

weak Leeuwin Current can be part of the reason for the excessively cold clumped-isotope and $\delta^{18}$O-derived calcification

temperatures obtained for MIS 96 (Fig. 5b-c).




**Figure 6: (a)** ODP Site 763 and IODP Site U1463 δ¹⁸O time-series from the Northern Carnarvon Basin (purple), compared to IODP Site U1459 δ¹⁸O time-series from the Perth Basin (orange). **(b)** Isotopic gradient between the Northern Carnarvon Basin (19°S) and the Perth Basin (29°S) along the Leeuwin Current pathway. **(c-d)** Forcing factors that are thought to influence Leeuwin Current intensity include eustatic sea level (dark green; Rohling et al., 2021) and the Summer (June 21st) Inter-Tropical Insolation Gradient (SITIG). They are combined in a composite record after normalizing. **(e-f)** Multi-taper method power spectrum of the isotopic gradient and forcing composite. Their spectral signature is similar, dominated by obliquity and precession, and without a strong 100-kyr imprint. Spectral peaks in (e) are not strongly expressed because the underlying records have been subject to conservative astronomical tuning and age-depth modelling.



### 4.2. Isotopic gradients along the Leeuwin Current pathway as a measure for poleward heat transport.

An effective way to investigate the temporal and spatial dynamics of the Leeuwin Current is by calculating proxy gradients at several localities along its pathway. In a previous paper, De Vleeschouwer et al. (2019) utilized the $\delta^{18}$O gradient between the Northern Carnarvon Basin (Site U1463 and 763) and the Perth Basin (Site U1459) to document mid-to-late Pliocene (3.9 – 3.1 Ma) Leeuwin Current dynamics. This paper extends the isotopic gradient approach across the Plio-Pleistocene boundary to 2.2 Ma (Fig. 6a, 6b). The extended analysis sheds new light on previous interpretations because, in the earlier work, a rapid increase in isotopic gradient (from 0.9 to 1.5‰, around 3.7 Ma) was interpreted as a permanent weakening of the Leeuwin current. It was claimed that this rapid change was likely caused by a tectonic reorganization of the Indonesian Throughflow. However, the extension of the isotopic gradient analysis presented here reveals that the steepening of the isotopic gradient was not permanent. Indeed, one observes a gradual return to flat isotopic gradients throughout the Late Pliocene (3.1 – 2.9 Ma, Fig 6b). By the latest Pliocene, between 2.9 – 2.6 Ma, latitudinal $\delta^{18}$O gradients were again as flat as 0.5‰ (Fig. 6b). Such flat gradient is indicative of a strong Leeuwin Current and was previously observed during the Early Pliocene, between 3.9 – 3.7 Ma. In other words, the rapid increase in isotopic gradient at 3.7 Ma and weakening of the Leeuwin Current were transient and are thus unlikely to have been caused by a tectonic reorganization in the throughflow region.

The new analysis corroborates other interpretations proposed in De Vleeschouwer et al. (2019): They argue that eustatic sea level and astronomical insolation forcing of wind patterns regulate Leeuwin Current dynamics on astronomical time scales. These two forcing factors are represented in Fig. 6c by token of the eustatic sea level reconstruction by Rohling et al. (2021) and the intertropical insolation gradient on June 21$^{st}$ (SITIG; Bosmans et al., 2015). Figures 6 and 7 were re-drawn with a vastly different sea level reconstruction in Suppl. Figs. B2-3 (reconstruction by De Boer et al., 2014), leading to the same result and interpretation. Eustatism plays a role because sea level lowstands weaken ITF connectivity and reduce the availability of shallow Leeuwin Current source waters. Wind patterns are important because the Leeuwin Current has to overcome strong southerly winds during austral summer but benefits from south-easterly winds blowing off the Australian continent during winter. When the SITIG index is maximum, the Hadley cell over Australia is enhanced during austral winter but weakened during summer. Both effects spur Leeuwin Current flow under maximum SITIG. The sea level and insolation series in Fig. 6c have been normalized and added to arrive at the composite signal in Fig. 6d. We thus propose a causal relationship between the combined forcing (Fig. 6d) and the isotopic gradient (Fig. 6b). This interpretation is endorsed by the broadly similar spectral character of both curves (Fig. 6e-f): Both spectra are dominated by obliquity and precession and do not exhibit a 100-kyr eccentricity peak. We emphasize that the obliquity and precession spectral peaks in Fig. 6e occur at the expected frequencies, even though the underlying age-depth models (Sites 763, U1463, and U1459) have been constructed using conservative tuning approaches. Indeed, none of those sequences have been tuned at the level of individual precession or obliquity cycles, with individual age-depth models not exceeding one tie point every ~100 kyr. The most compelling argument for the proposed causal connection comes from the remarkable co-variation between forcing and isotopic gradient on astronomical timescales,



displayed in Figure 7. Different glacial stages are recognizable in the forcing composite series and the isotopic gradient:
Intervals with a low forcing are generally characterized by a steep isotopic gradient and thus a weak Leeuwin Current. Also,

the relative strength of individual glacial-interglacial cycle is consistent between forcing composite and isotopic gradient. The
two curves are largely concurrent on longer timescales, notably between 3.1 – 2.2 Ma and before 3.7 Ma. However, between
3.7 and 3.1 Ma, the Leeuwin Current isotopic gradient is steeper than what one would derive from the similarity between
forcing and gradient in the rest of the record (Fig. 7). We infer a prolonged period of Leeuwin Current weakness during that
time. Its timing chiefly corresponds to a time of Southern Hemisphere cooling, marked by the oldest tills in Argentinian

Patagonia (3.6 Ma; Clague et al., 2020), a period of continental aridity in the south-central Andes (3.6 - 3.3 Ma; Amidon et al.,
2017), the absence of the *Dictyocha* silicoflagellate and low abundance of calcareous nannoplankton between 3.5 – 3.2 Ma at
ODP Site 751 on the Kerguelen Plateau (Bohaty and Harwood, 1998), a cooling of more than 3°C between 3.7 – 3.5 Ma based
on warm vs. cold silicoflagellates from ODP Site 1165 on the Prydz Bay continental rise (Escutia et al., 2009), a marked
decrease in Sub-Antarctic Zone diatoms because of surface ocean water cooling at DSDP Site 513 in the Atlantic sector of the

Southern Ocean (3.6 Ma; Kato, 2020), and a gradual increasing trend in seasonal sea-ice indicating diatoms at ODP Site 689
in the Weddell Sea (starting at 3.7 Ma; Kato, 2020).

The temporal evolution of the Leeuwin Current isotopic gradient strongly resembles alkenone-based SST reconstructions from
the wider Agulhas region (South Atlantic, Fig. 8), with a general cooling trend between 3.7 – 3.1 Ma and almost stable to

slowly increasing temperatures between 3.1 Ma and the earliest Pleistocene (Site 1090 from Martínez-Garcia et al., 2010; Site
1088 from Herbert et al., 2016; Site 1087 from Petrick et al., 2018). The SST evolutions of those sites are coupled in varying
degrees to latitudinal shifts in sub-Antarctic climate belts. The synchronicity between Leeuwin Current intensity and Agulhas
SSTs thus suggests that there is a mechanistic link between the latitudinal position of the subpolar frontal system and Leeuwin
Current intensity. This also implies that the rapid steepening of isotopic gradients at ~3.7 Ma was probably associated with an

intensified SAMW production in the Indian Ocean and possibly an equatorward shift of Southern Hemisphere climate belts
rather than a tectonic event in the ITF region. Despite the co-variation, open questions remain as to the exact cause and effect.
It is rather unlikely that Leeuwin Current weakening would be the direct effect of an equatorward shift of the subtropical front.
That is because such change would steepen the annual mean steric height gradient alongshore Western Australia, which would
result in strengthening, not weakening, of the Leeuwin Current. The other way around, Karas et al. (2011) suggested that a

dwindling surface ITF has important implications for the Leeuwin Current and the poleward heat flux, possibly leading to a
cooling of the Benguela upwelling system. Building on that suggestion, De Vleeschouwer et al. (2018) described a positive
feedback mechanism for Pliocene Southern Hemisphere climate change: An initial reduction in ITF heat transport leads to
(sub-)Antarctic cooling, further glacio-eustatic sea level fall, and more Leeuwin Current weakening. Based on these
hypotheses, it can be speculated that a reduction in poleward heat transport through the Leeuwin Current and the Agulhas

(Return) Current caused the observed Southern Hemisphere cooling between 3.7 – 3.1 Ma. However, in this scenario, the
question remains as to what triggered the strong cutback in poleward heat transport at ~3.7 Ma in the first place. Possibly, the



relatively large sea level drop related to the Early/Late Pliocene expansion of Northern Hemisphere ice sheets (De Schepper et al., 2014) was the trigger for the reduction in Indian Ocean (Leeuwin and Agulhas current) poleward heat transport. This initial change could have been amplified through the positive feedback mechanism in which the reduction of Indian Ocean

poleward heat transport advances the thermal isolation of Antarctica (Karas et al., 2011; De Vleeschouwer et al., 2018), bringing the Southern Hemisphere into a long-lasting "cool state".

On the other hand, $TEX_{86}$ and clumped-isotope paleotemperatures demonstrate that the Leeuwin Current continued to operate throughout the Pliocene, even when the latitudinal isotopic gradient became steeper. These new datasets thus provide important

constraints for further ocean and climate modelling that can test the remaining open questions: Were Early/Late Pliocene sea level eustatic lowstands capable of sufficiently reducing Indian Ocean poleward heat transport so that it could influence Southern Hemisphere climate (without completely halting the Leeuwin Current)? Or were reductions in Leeuwin Current intensity the (indirect) result of an equatorward expansion of the sub-Antarctic climate belts, despite the corresponding steepening of the alongshore steric height gradient?






**Figure 7**: **(a)** Comparison of the isotopic gradient and forcing composite reveals co-variation on astronomical timescales. **(b)** Comparison of clumped isotope calcification temperatures and forcing composite. The absence of temperature change between different samples indicates that the Leeuwin Current remained active all throughout the studied interval, despite variability in Leeuwin Current intensity. Sample PB06 is, however, much colder than others, which we in part explain by upwelling of cold sub-Antarctic mode waters under a weak Leeuwin Current regime during MIS 96 (2.44 Ma) as indicated by an increase in U/Th around that time (Fig. 5c).



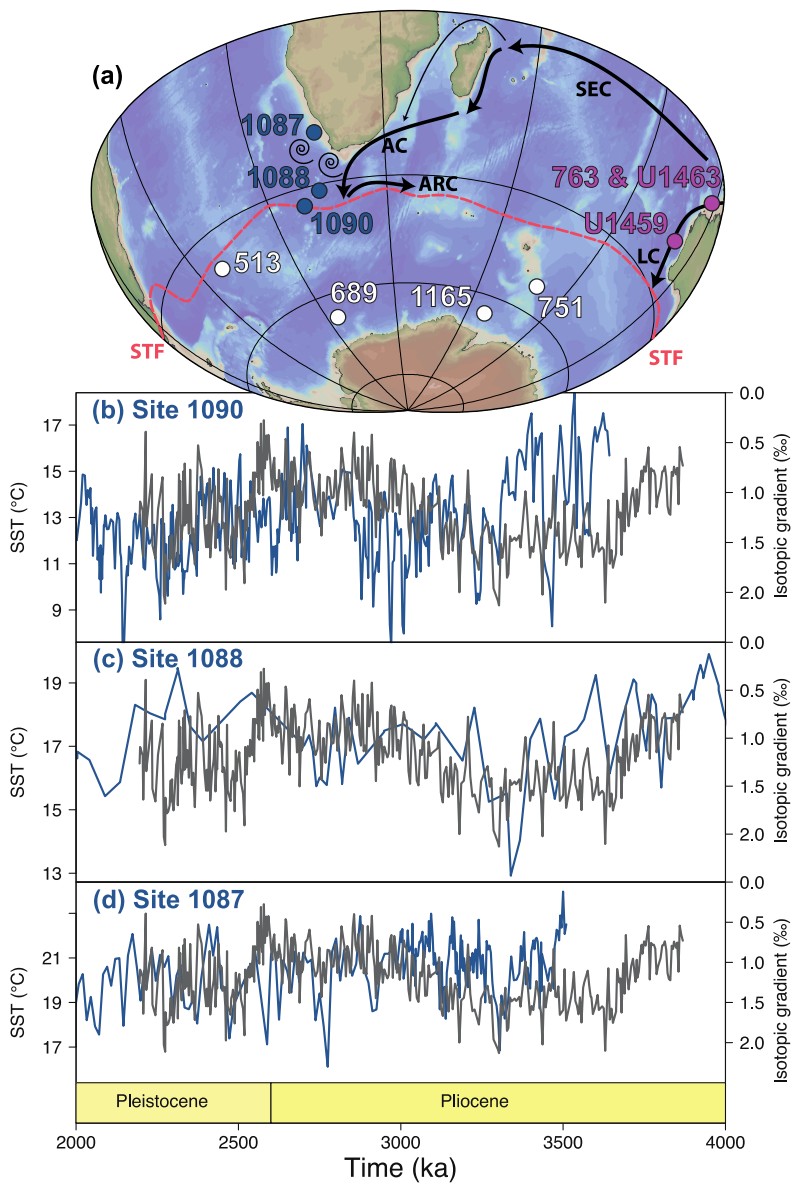

**Figure 8: (a)** Map of the South Atlantic, Indian and Southern Oceans indicating the location of DSDP, ODP, and IODP Sites discussed in this work. LC = Leeuwin Current; SEC = South Equatorial Current; AC = Agulhas Current; ARC = Agulhas Return Current. **(b-d)** Comparison of the Leeuwin Current isotopic gradient (dark grey) and different alkenone-based SST reconstructions (blue) in the Agulhas/Benguela region (Site 1090 from Martínez-Garcia et al., 2010; Site 1088 from Herbert et al., 2016; Site 1087 from Petrick et al., 2018).



## 5 Conclusions

We have reconstructed a Plio-Pleistocene history of Leeuwin Current dynamics and Perth Basin ocean water temperatures, using a multiproxy approach at IODP Site U1459. Clumped isotope and TEX$_{86}$-derived paleotemperatures for Site U1459 are about 5°C apart, which we explain in the first place by a difference in the exact water depth that is represented by both proxies.

TEX$_{86}$ likely reflects true surface temperatures, albeit with a possible seasonal warm bias. Clumped isotopes measured on *T. sacculifer* probably mirror calcification temperatures from the lower mixed layer or upper thermocline. We reconstruct Leeuwin Current intensity by calculating δ$^{18}$O differences along a 19-29°S latitudinal transect.

This analysis confirmed the dependency of Leeuwin Current intensity on eustatic sea level and insolation forcing of wind

patterns. Moreover, it indicates a "weaker than expected" Leeuwin Current between 3.7 – 3.1 Ma. This time interval chiefly corresponds to cool climate conditions across the Southern Hemisphere and a more northerly position of the subtropical front. The Leeuwin Current isotopic gradient exhibits remarkable congruity with SST records from the Southern Atlantic Ocean, supporting earlier proposed links between Leeuwin Current intensity and oceanographic change in the Agulhas / Benguela region. While this result underlines the importance of Indian Ocean poleward heat transport through the Leeuwin Current and

the Agulhas (Return) Current, it remains an open question why the Leeuwin Current remained "weaker than expected" for ~600 kyr after 3.7 Ma. Finally, the coolest calcification temperatures (δ$^{18}$O and Δ$_{47}$) at Site U1459 were obtained for MIS 96, which may represent the upwelling of sub-Antarctic Mode waters onto the Australian shelf.





**Appendix A: Age-depth models of IODP Site U1459 and ODP Site 763**

| U1459 Depth (mcd) | Age (ka) |
|---|---|
| 51.42 | 1930.00 |
| 56.71 | 2262.00 |
| 60.35 | 2365.00 |
| 64.37 | 2508.79 |
| 67.65 | 2584.03 |
| 73.03 | 2803.77 |
| 80.55 | 3015.00 |
| 84.02 | 3134.44 |
| 87.79 | 3302.81 |
| 99.37 | 3669.92 |
| 109.83 | 3965.10 |
| 117.67 | 4157.35 |
| 119.79 | 4230.30 |
| 123.95 | 4345.25 |
| 153.43 | 5128.20 |
| 155.66 | 5216.66 |
| 159.04 | 5380.93 |
| 161.04 | 5456.61 |

**Table A1:** Age-depth model for the Plio-Pleistocene interval of IODP Site U1459

| 763 age (Karas et al., 2011) | 763 age (this study) |
|---|---|
| 2019 | 2032 |
| 2414 | 2463 |
| 2708 | 2716 |
| 2967 | 2975 |
| 4000 | 4000 |
| 6000 | 6000 |

**Table A2:** A minor chronology adjustment of Site 763 compared to Karas et al. (2011) was needed in the youngest part of the record. This is because in the interval younger than 3.1 Ma, the Karas et al. (2011) age-depth model is solely based on magnetic reversals. The minor chronology adjustment shown here was necessary to make sure that heavy planktonic isotope
compositions correspond to glacials and vice versa.





**Appendix B: Present-day oceanography of the south-eastern Indian Ocean**



**Figure B1: Present-day oceanography of the south-eastern Indian Ocean.** The southward flowing Leeuwin Current is strongest in late autumn and winter. During that season, the Leeuwin Current forms large eddies, causing significant deepening
of the mixed layer (up to ~200 m) and facilitating enhanced primary productivity (shown here: data from July 1st, 2021). During austral summer, the Leeuwin Current is weaker (shown here: January 1st, 2021), with a thinner mixed layer and reduced productivity is reduced. Generated using E.U. Copernicus Marine Service Information with GLOBAL_ANALYSIS_FORECAST_PHY_001_024 (Zammit-Mangion and Wikle, 2020) and GLOBAL_ANALYSIS_FORECAST_BIO_001_028.






**Appendix C: *T. sacculifer* scanning electron microscope images**

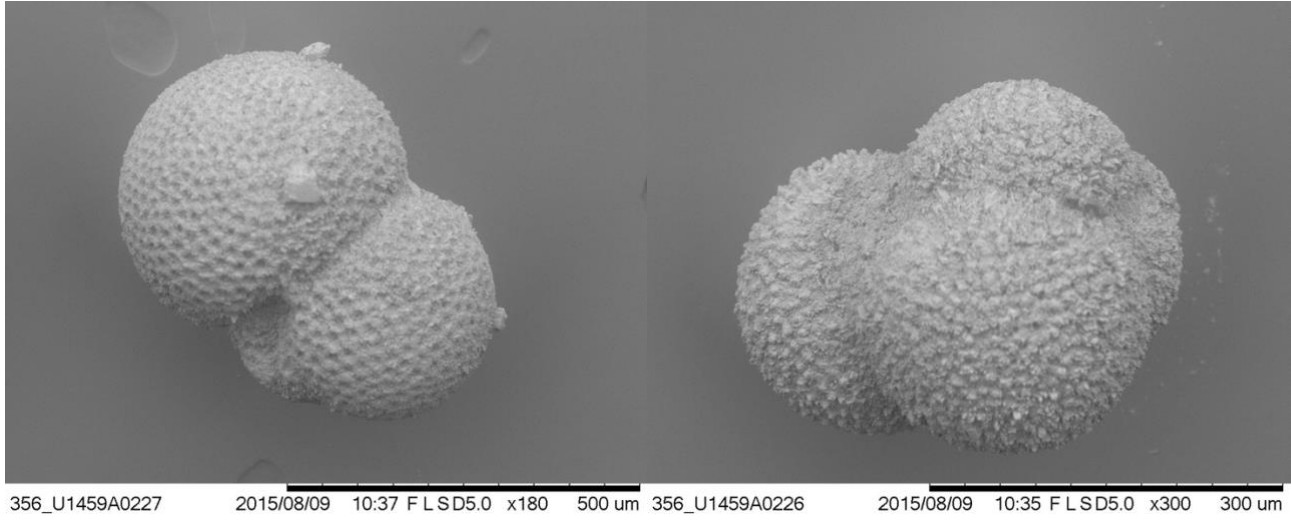

**Figure C1: Representative SEM images of *T. sacculifer*,** illustrating their preservation (Gallagher et al., 2017)






## Appendix D: Alternative sea level reconstruction

**Figure D1:** Same Figure as Figure 6 in the main manuscript, but showing the sea level reconstruction of De Boer et al. (2014) in panel (c). This sea level reconstruction is then used to calculate the forcing composite in panel (d). The key message here is

that the choice of sea level reconstruction does not influence the interpretations made in this study.





**Figure D2:** Same Figure as Figure 7 in the main manuscript, but using the sea level reconstruction of De Boer et al. (2014) to calculate the forcing composite. The key message here is that the choice of sea level reconstruction does not influence the interpretations made in this study.



**Code availability:** The R code used to produce Figures 3 through 8 is available on Zenodo:
https://doi.org/10.5281/zenodo.5638746

**Data availability:** All Site U1459 stable isotope data will be permanently archived through PANGAEA https://doi.pangaea.de/10.1594/PANGAEA.XXXXXX. For the period of peer review, reviewers can access this data through the Zenodo link in the code availability statement; The Site U1459 XRF and TEX$_{86}$ data is available at 580 https://doi.pangaea.de/10.1594/PANGAEA.903102 ; The Site U1463 stable isotope dataset is available at https://doi.pangaea.de/10.1594/PANGAEA.892422 ; The Site 763 stable isotope dataset is available at ftp://ftp.ncdc.noaa.gov/pub/data/paleo/contributions_by_author/karas2011/karas2011.txt .

**Author contribution:** DDV conceived and directed the study; AF washed and sieved sediment samples, and picked the 585 foraminifera for isotopic analysis; AF and NM carried out sample preparation; MP and MM carried out clumped isotope measurements in the Brussels lab, in consultation with SG, CS and PC. BP carried out the TEX$_{86}$ measurements at the Max Planck Institute for Chemistry in Mainz. DDV, MP, NM, SG, CS and PC contributed to the interpretation and discussion of the results. DDV wrote the paper with input of all authors.

**Competing interests:** The authors declare that they have no conflict of interest.

**Acknowledgements:** This research used samples and data provided by the International Ocean Discovery Program (IODP). We thank all IODP Expedition 356 scientific participants and crew for making this study possible. The VOCATIO Foundation provided funding through a scholarship to DDV (2016 promotion). At the time of writing, DDV was a senior scientist funded 595 through the Cluster of Excellence "*The Ocean Floor - Earth's Uncharted Interface*" (German Science Foundation, DFG). The Max Plank Society provided funding through a postdoctoral grant to BP. We thank Kara Bogus for practical support during XRF scanning, Gerald Auer for insightful discussions, and Aisling Dolan and Caroline Prescott for providing the Pliocene HadCM3 simulations shown in Figure 2. Research Foundation Flanders (FWO) Hercules funding supported the purchase of the IRMS. PC acknowledges the VUB Strategic Research Program




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
