# Peer review of "Plio-Pleistocene Perth Basin water temperatures and Leeuwin Current dynamics (Indian Ocean) derived from oxygen and clumped isotope paleothermometry"

_Climate of the Past, 2021_

## Author Comment (AC1)

**Final Author Comment**

De Vleeschouwer et al. – cp-2021-151

We thank referee R3 for their in-depth and constructive assessment of our manuscript. The first major comment refers to the TEX$_{86}$ – clumped isotope discrepancy and we fully agree with the reviewer in this matter. The second comment contains several questions as to which environmental factors could force the isotopic gradient. To answer these questions, we present new arguments that indicate mixed-layer temperature gradient along the 19-29°S transect as the dominant driver. The third comment raises some legitimate concerns about clumped isotope statistics. In answer to these concerns, we justify certain choices and follow the reviewer in most of their requests. The fourth comment relates to the clumped isotope calibration curve by Peral et al. (2018): The recalculation of this calibration will be published in the next couple of weeks by *Geochimica et Cosmochimica Acta* (GCA). All minor comments raised by referee #3 are adopted.

**1. TEX$_{86}$ vs. clumped temperatures**

*Firstly, I have some concerns about the authors' conclusions that TEX86 likely reflects sea surface temperatures while the clumped isotope analyses are indicative of the temperatures in the lower mixed layer, as put forward in the abstract (lines 24-26). In their discussion about the discrepancy between TEX86 and clumped temperatures, the authors put forward the hypothesis that TEX86 may be seasonally biased (lines 343-350). In fact, previous studies have suggested that a summer bias on TEX86 is likely and explains the consistent difference between TEX86 and stable isotope paleotemperature estimates found in other studies (Jia et al., 2017; O'Brien et al., 2017). While they disregard their other working hypothesis about the southward displacement of sinking particles, it seems that the authors cannot exclude the possibility of seasonal bias in their TEX86 data (see line 387), which could easily explain the ~5°C difference between TEX86 and clumped results (line 283) if TEX86 represents summer SST and clumped records MAT (see Fig. 1). In absence of clear evidence about the living habitat of T. sacculifer (lines 371-385), and with the only other line of evidence being that the TEX86 temperatures "seem reasonable compared to the present-day mean annual temperatures" (lines 341-342), I think the conclusion that clumped isotope temperatures represent the lower mixed layer and TEX86 represents the mean annual SST is not sufficiently supported.*

The reviewer is absolutely right: We cannot exclude the possibility of a seasonal warm-bias in the TEX$_{86}$ data. In fact, there is good evidence that a seasonal warm bias is affecting our TEX$_{86}$ data from Site U1459. In lines 345 – 347, we already indicated that observations by Smith et al. (2020) and Benjamin Petrick (pers. comm.) at respectively Sites U1463 and U1460 suggest a ~3°C warm bias using the BAYSPAR surface temperature calibration (see Fig. S6 in Smith et al., 2020, where core-top TEX$_{86}$ temperatures are systematically higher than mean annual temperatures at those localities). This warm bias more than likely reflects a seasonal bias, as the observed ~3°C difference roughly corresponds to the difference between present-day summer sea surface temperature (~25°C) and mean annual sea surface temperature (~22.8°C). Hence, we are on the same page with the reviewer. We admit that lines 24-26 in the abstract are short on detail, and that we shouldn't have labeled this seasonal warm bias as "minor" in line 357. This will be fixed.

However, a ~3°C warm bias on the TEX$_{86}$ temperatures does not fully resolve the discrepancy with the clumped isotope temperatures. A systematic difference of ~2°C remains. The reported clumped-isotope calcification temperatures range between 18.2 – 20.8°C, suggesting a lower mixed layer habitat depth for *G. sacculifer*. On the one hand, we will elaborate on a possible cold-bias on the clumped temperatures in §4.1 in response to comments by Clara Bolton (R1) and R2. On the other hand, we will emphasize that strategies were implemented to minimize the clumped-isotope cold-bias (avoiding partly dissolved specimens and cleaning protocol to remove microcrystalline sparitic cements).

**2. Isotopic gradients**

*While I understand that the authors only measured clumped isotope temperatures in one site, it is a shame that their discussion of the isotopic gradient along the Leeuwin Current does not benefit from the addition of clumped isotope analyses. My very first question on reading this discussion after the discussion on the clumped isotope results is how much of this isotopic gradient reflects temperature gradient and how much reflects the difference in seawater isotopic composition. Would a strengthening or weakening of the Leeuwin Current affect both these variables similarly? Would it somehow be possible to infer from the changes in temperature and seawater oxygen isotope composition over time, which the authors can infer from their clumped isotope record, whether the changes observed in δ18O over time are mostly driven by temperature or water composition? And by extension, could this evidence be used to say something about which factor predominantly forced the changes in isotopic gradient? Finally, if the author's hypothesis that the foraminifera calcify in the lower mixed layer is correct (see previous comment), how does this impact the discussion of isotopic gradient? Can the authors somehow exclude that changes in the calcification depth or the depth of the mixed layer between the two sites which are compared affect the difference in δ18O without the need for a change in the strength of the Leeuwin Current? I feel that there is some untapped opportunity for discussion on this topic which would integrate the clumped isotope analyses more firmly into the main discussion of the manuscript.*

This question combines two comments made by Clara Bolton (R1) and R2:

a)  In response to Clara Bolton, we provide an estimate of salinity changes on G-IG timescales. We infer a maximum of 0.6 psu salinity difference between phases of strong Leeuwin Current (Interglacial, present-day analogue: austral winter) and weak Leeuwin Current (Glacial, present-day analogue: austral summer). This estimate of salinity change on G-IG timescales is based on the present-day seasonal variability and likely represents an over-estimation because the Leeuwin Current never shut down completely during glacial intervals. Translated to $\delta^{18}O_{sw}$, this corresponds to 0.10 – 0.15‰, which is minor compared to the variability in the isotopic gradient (2‰).

b)  In response to R2, we refer to the results of He et al. (2021). The black line in the plot below (i.e. Fig. 3C in He et al., 2021) shows the $TEX_{86}$ gradient between Sites U1461 (also in the northern Carnarvon Basin, just as Sites U1463 and 763) and U1459 (Perth Basin). The low-resolution reconstruction of Leeuwin Current intensity by He et al. (2021) is in excellent agreement with our high-resolution $\delta^{18}O$-based reconstruction (orange line). As the $TEX_{86}$ gradient is solely temperature-driven, the excellent co-variation provides strong support for interpreting the isotopic gradient as mainly temperature-driven. This additional line of argumentation will be implemented in the revised version of the manuscript.

Based on the above points, we make the case that the mixed-layer temperature gradient between 19 – 29°S, in response to Leeuwin Current strength, is the factor predominantly forcing the reported changes in isotopic gradient. We consider changes in the calcification depth or the depth of the mixed layer between the two end-members unlikely, as this would cause changes in the isotopic gradient that are not observed in the $TEX_{86}$ gradient.

[Figure]

Comparison of the TEX86 gradient between Sites U1461 and U1459 (Fig. 3C in He et al., 2021) and the isotopic gradient from our work. Both proxies are completely independent of each other, yet show similar patterns throughout the Plio-Pleistocene, indicating the robustness of these results.

**3. Clumped isotope statistics.**

*First of all, the caption of Table 2 (line 309) and the methods description (line 277) list different reproducibility errors for the clumped isotope measurements. I assume that the standard deviations cited in line 277 are one order of magnitude too high (e.g. 0.0314‰ instead of 0.314‰, as in line 309).*

Indeed: Line 277 should report 0.0314 and 0.0173‰. This will be corrected.

*Secondly, I noticed that the authors used the reproducibility of their standards for calculating the standard errors in Table 2 (see lines 307-309) instead of the within-sample reproducibility. This method is likely to underestimate the uncertainty on the D47 values in the samples, as the homogenized ETH-4 standard on which the standard deviation is based will likely reproduce better than the samples consisting of foraminifera pooled from up to four adjacent samples (line 189; up to 60 cm core depth when using the median sampling resolution from line 177). The authors should at least report the reproducibility of clumped isotope analyses within their samples.*

The reviewer is correct: because of the low number of repeated measurements on samples, we chose to use the standard deviation on the standards to calculate the standard error. Our reasoning was that, with just a handful of repeated measurements on hand, using the sample standard deviation to calculate the standard error would be a case of *small number statistics*. We would like to point out that the measurement strategy with the NuCarb, compared to the more common MAT253+ setup, is such that relatively few repeated measurements are made with relatively large homogenized samples (500 micrograms vs. 80 - 120 micrograms). For this reason, in this particular case, we considered that the standard error on our sample measurements was best estimated by calculating the standard error from the standard deviation on the standards.

That being said, we will implement the reviewer's suggestion to report the reproducibility of individual sample clumped isotope analyses in an additional column added to Table 2. Furthermore, we will move away from solely using the standard deviation on standards to calculate

the standard error, and instead we will adopt the full propagation of analytical uncertainties in Δ47 measurements as described by Daëron (2021). He proposes to adopt a "pooled" standardization method that considers constraints on reproducibility available from both standard and sample analyses. This approach has been shown to yield realistic error estimates. In the text, we will add a statement as to why the "pooled" approach is the best way forward to estimate standard errors in this case with relatively small numbers of replicate measurements.

Moreover, additional aliquots of the eight clumped isotope temperatures have been measured to reduce the error bars on those temperatures. These will be included in the revised manuscript.

*Thirdly, in the clumped community it is common practice to report uncertainties at the 95% confidence level (e.g. Fernandez et al., 2017). Instead, the authors report uncertainties at ±1 standard error in Tables 2 and 3. The captions of Figures 4 and 5 do not show what the error bars on the clumped datapoints represent, but from comparison with the tables I infer that these are also 1 SE. This reporting makes the uncertainty look smaller than in other studies using 95% confidence level and in my opinion the reporting of ±1 SE ("68% CL") is less intuitive. I realize that calculating 95% CL, or even the within sample standard deviation, of samples with 2 or 3 replicates (PB03, PB05, PB06 and PB08) is challenging due to the lack of statistics. This problem illustrates the risk of analyzing small numbers of replicates of samples and will make it challenging to assess the confidence on these clumped isotope datapoints, or to compare the results amongst themselves (e.g. via a Student's T-test) or with other data. I do not know how this issue can be resolved without adding additional replicates, and I do sympathize with the authors given how much work it is to gather enough foraminifera for these measurements. At the very least, I would therefore urge the authors to add information about their within sample reproducibility (standard deviations) for all samples and calculate 95% confidence levels for those samples for which this is feasible (sample size > 3), in addition to making the clumped isotope results available in an open-access repository (now, only regular stable isotope data is archived).*

In the revised manuscript we will include additional measurements, such that all clumped temperatures will be based on a sample size > 5. Following the suggestion of the reviewer, we will show both the 68 and 95% confidence levels in the figures, and the 95% confidence levels in Tables 2 and 3. All figure captions will clearly indicate that the 68% and 95% confidence levels are shown, and that this uncertainty is randomly distributed. For the sake of consistency, we will also show the 95% confidence interval (2.5% - 97.5%) on the TEX$_{86}$ temperatures in Fig. 4

The clumped isotope final results (incl. temperatures) will be made available in the PANGAEA open-access repository upon publication of the manuscript, together with all other proxy data. Moreover, we will upload the raw clumped data (including machine data and correction steps) to the EarthChem repository. This way, EarthChem holds all data to allow future clumped experts to recalculate any data.

*Finally, while not (yet) a standard in the clumped isotope community, it would be good practice if the uncertainty on the clumped isotope calibration(s) used in the study were to be propagated on the clumped isotope result. This uncertainty is not contained within the measurement uncertainty and is usually relatively small (<5 ppm). However, given the differences between the sample sizes and temperature ranges between the calibrations cited in Table 3, the differences in uncertainties of these calibrations could be discussed.*

Ucertainties on the calibrations will be propagated to the temperature results in the revised version of the manuscript.

**4. Recalculated clumped isotope calibration**

*It is a really nice addition that the authors compare the results of applying difference clumped isotope calibrations on their data (Table 3). This gives the reader a good intuition of the difference using different calibration makes in the study. Unfortunately, the I-CDES scale updated calibration of Peral et al. (2018) is not provided in the paper. This makes it impossible to verify the temperature results of this calibration. For the sake of open science, I urge the authors to make the I-CDES-scale referenced calibration dataset on which the updated Peral et al. (2018) calibration is based available in an open repository (e.g. Pangaea or EarthChem database) and to cite the new calibration formula with uncertainties on slopes and intercepts (sensu Equation 1) in the manuscript text or in Table 1 (as in Meinicke et al., 2021). Providing the calibreation dataset is especially important as the uncertainties on the calibration (see previous comment) cannot be propagated from the errors on the slopes and intercepts of the calibration formula alone, as information about the covariation of slope and intercept are missing from this information.*

The I-CDES scale updated calibration of Peral et al. (2018) is currently under review with *Geochemica Cosmochemica Acta* and is expected to be accepted in the coming weeks. This anticipated work Peral et al. (2022) will contain the calibration dataset and will be cited in our manuscript as we did with Meinicke et al. (2021) for the Meinicke et al. (2020) calibration.

We will add the uncertainties on slopes and intercepts on all calibrations in Table 1.

**5. Minor Comments**
- Line 24-26, 39, 195, 451, 535 will be rephrased
- Line 457: This is an excellent suggestion: cross-spectral analysis will be added to underpin the co-variation between different time-series on different timescales.
- Figure 7: ("permille") will be added to the isotopic gradient axis, and "(unitless)" will be added to the composite axis.
- Figure C1: Pictures are from Gallagher et al. (2017): This will be clarified.

Daëron, M. (2021). Full Propagation of Analytical Uncertainties in Δ47 Measurements. *Geochemistry, Geophysics, Geosystems, 22*(5), e2020GC009592. https://doi.org/10.1029/2020GC009592

Gallagher, S. J., et al. (2017). Site U1459. In S. J. Gallagher, C. S. Fulthorpe, K. Bogus, & the Expedition 356 Scientists (Eds.), *Indonesian Throughflow.* (Vol. 356). College Station, TX: Proceedings of the International Ocean Discovery Program.

He, Y., et al. (2021). Development of the Leeuwin Current on the northwest shelf of Australia through the Pliocene-Pleistocene period. *Earth and Planetary Science Letters, 559*, 116767. https://doi.org/10.1016/j.epsl.2021.116767

Meinicke, N., et al. (2020). A robust calibration of the clumped isotopes to temperature relationship for foraminifers. *Geochimica et Cosmochimica Acta, 270*, 160-183. https://doi.org/10.1016/j.gca.2019.11.022

Meinicke, N., et al. (2021). Coupled Mg/Ca and Clumped Isotope Measurements Indicate Lack of Substantial Mixed Layer Cooling in the Western Pacific Warm Pool During the Last ~5 Million Years. *Paleoceanography and Paleoclimatology, 36*(8), e2020PA004115. https://doi.org/10.1029/2020PA004115

Peral, M., et al. (2018). Updated calibration of the clumped isotope thermometer in planktonic and benthic foraminifera. *Geochimica et Cosmochimica Acta, 239*, 1-16. https://doi.org/10.1016/j.gca.2018.07.016

Smith, R. A., et al. (2020). Plio–Pleistocene Indonesian Throughflow variability drove Eastern Indian Ocean sea surface temperatures. *Paleoceanography and Paleoclimatology, 35*(10), e2020PA003872. 10.1029/2020pa003872

---

## Author Comment (AC2)

**Final Author Comment**

De Vleeschouwer et al. – cp-2021-151

We thank R2 for their assessment of our manuscript. The first major comment demands for a more comprehensive discussion of the effects of dolomitization, dissolution, and recrystallization on stable isotope measurements and their paleoclimate interpretations. The second, third and fourth major comment all refer to the concept of using isotopic gradients along the flow path of the Leeuwin Current as a proxy for its intensity. The skepticism and unclarities brought up by the reviewer can be addressed in full. Most detailed comments will also be addressed.

**1. The impact of diagenesis on stable isotope results**

*A planktonic d18O record is presented from core depths (55-105 mcd) where partly severe dolomitization was reported previously (Proceedings of the International Ocean Discovery Program, vol. 356). Further, all samples are from a shallow carbonate rich ocean region close to coral reefs where carbonate diagenesis is very common. It is well known that diagenesis and recrystallisation at sea bottom alter foraminiferal d18O towards heavier values (e.g., Edgar et al., 2015, Geochimica et Cosmochimica Acta). It is also clear that these alterations (solution, recrystallisation) are not always visible in the crystal structure of the foraminifers (Kozdon et al., 2011, and refs. therein, Paleoceanography). The comparison to clumped isotope temperatures is not convincing to rule out diagenesis as there are only eight data points shown and it has been shown that clumped isotope temperatures from planktonic foraminifers are also biased towards colder temperatures by diagenesis (Leutert et al., 2019, Geochimica et Cosmochimica Acta). Hence, at least parts of the paleoclimatic interpretations with relatively high d18O during the warm Pliocene might be probably related to diagenesis.*

We agree with the reviewer that our temperature reconstructions are not to be considered 100% reliable in terms of their absolute values. But when are they ever (Cisneros-Lazaro et al., 2022)?

Dolomite indeed constitutes between 2 – 24% of the dominant mineral phases in the studied portion of Site U1459 (based on XRD analyses reported in Table T6 of the U1459 Site Report in the Proceedings of the International Ocean Discovery Program, vol. 356). We infer this is cryptocrystalline dolomite as dolomite was not observed in smear slides between 0 – 120 m CSF-A (Figure F5 in the U1459 Site Report). A similar pattern was observed in terms of planktonic foraminifera, with dolomite crystallization affecting foraminiferal specimens at cored depths >136.9 m CSF-A (p. 19 in U1459 Site Report). Dolomite is thus recognized throughout Site U1459 and becomes a stronger influence downcore. However, we do not observe a systematic change towards more positive $\delta^{18}O$ values with increasing core depth. On the contrary, **our data behaves opposite to the expected diagenetic imprint in high-carbonate settings** (e.g., Stainbank et al., 2020). We thus conclude that dolomite did not impair the planktonic foraminifera as paleoclimate recorders in the studied interval.

The possible cold-bias due to diagenesis (recrystallization in particular) has also been raised by Clara Bolton (R1) and we recognize that a diagenetic cold-bias deserves a more extensive discussion in the revised version of the manuscript. Specifically, a cold-bias paragraph will be added to §4.1. Additionally, in §2.3, we will stipulate that, while foraminifera were generally of good preservation, some specimens were partly dissolved and filled with thin layer of microcrystalline sparitic cement. The revised manuscript will mention that partly-dissolved specimens were dismissed for geochemistry, and a cleaning protocol has been in place to remove secondary calcite prior to clumped isotope analysis.

**2. Comments on the reliability of isotopic gradients for the reconstruction of Leeuwin Current strength**

*I see the method of calculating planktonic foraminiferal gradients to reconstruct changes in the paleo-Leeuwin Current (warmer-colder) very problematic. It is known that the d18O of planktonic foraminifers are dependent on local temperature changes, local salinity changes and the global ice volume. Even if the global ice volume is known from the past there are still two variables which are unknown for each site location (local temperature and local salinity). Also, the clumped isotope temperatures do not really support the presented d18O record from Site 1459 as a temperature signal. This is due to the very few (eight) data points over the whole time period studied, that makes it impossible to compare long-term trends in temperature. Additionally, these data points show a huge error bar of up to 10°C.*

This comment is rooted in an important misunderstanding. The method of calculating planktonic foraminiferal gradients is **designed to reconstruct changes in Leeuwin Current strength, not Leeuwin Current temperature**. This method starts from the assumption that -when the Leeuwin Current was **strong**- both endmembers of the 29°S-19°S transect were essentially bathed in the same water mass. When the Leeuwin Current was **weak**, the southern endmember (Site U1459) is expected to experience enhanced local cooling and local salinization of mixed layer waters compared to the localities of the northern endmember (Site U1463 and Site 763). In the latter *weak Leeuwin Current* case, Site U1459 undergoes a stronger shift towards more positive mixed-layer $\delta^{18}O$ values than Site U1463 / Site 763, and the isotopic gradient steepens. To avoid this misunderstanding among future readers, we will rewrite and extend the final paragraph of the introduction.

Additional aliquots of the eight clumped isotope temperatures have been measured to reduce the error bars on those temperatures. These will be included in the revised manuscript.

*It was not clear to me why the d18O gradient between sites at 29° S and 19° S reflect the evolution of the Leeuwin Current better than the difference between sites located northwards (sites 1463 and 763). Presented model simulations (Fig. 2b) show miniscule temperature changes at about 29° S at Site 1459 between cold and warm stages but the gradients presented by the authors are mostly driven by huge changes in the planktonic foraminiferal d18O of Site 1459.*

The isotopic difference between the northwards located sites (Site U1463 and 763) has been discussed and interpreted in De Vleeschouwer et al. (2018, EPSL, Fig 5D therein, see below). Basically, we found negligible differences in the isotopic values between both Sites throughout the studied Pliocene interval, with the exception for glacial Marine Isotope Stage M2 (MIS M2 at ~3.3 Ma). The indistinguishable $\delta^{18}O$ values at Sites U1463 and 763 indicate that both sites were bathing in the same water mass throughout the studied interval. Except for MIS M2 of course, which is when the Leeuwin Current reached its weakest Pliocene intensity. This low in Leeuwin Current strength caused Site 763A to temporarily reflect an Indian Ocean, rather than an ITF signal. For a more detailed discussion of these results, we refer the reader to De Vleeschouwer et al. (2018).

The reviewer's suggestion to look at the U1463-763 isotopic gradient to reveal Indian Ocean dynamics thus only works when the Leeuwin Current is exceptionally weak (like during MIS M2). For most of the Pliocene and early Pleistocene, though, the U1463-763 gradient is too small-scale to reveal ocean dynamics on glacial-interglacial (G-IG) timescales. Therefore, to reconstruct variations in Leeuwin Current strength and ITF connectivity on G-IG timescales, one needs to look at larger-scale gradients, over longer distances, for example, Perth Basin vs Carnarvon Basin.

[Figure]

Figure 5A-D from De Vleeschouwer et al. (2018): This panel shows identical $\delta^{18}O$ values at Site U1463 and Site 763 throughout most of the studied interval, except for MIS M2.

Planktonic foraminiferal $\delta^{18}O$ of Site U1459 varies between -0.5 and +0.5 permille, which is not "*huge*" nor does the U1459 $\delta^{18}O$ overwhelm the isotopic gradient. We believe that this is sufficiently clear from Fig. 6a-b. Hence, no concrete changes to the manuscript are planned in response to this comment.

The numerical climate model simulations in Fig. 2 are included in the manuscript to illustrate a mismatch between state-of-the-art Pliocene climate simulations and proxy data. The mismatch is likely the result of the coarse spatial resolution of climate models, thereby failing to capture detailed yet important paleoceanographic dynamics. The discrepancy acts as an additional motivation for our study, which is indicated in the introduction and the figure caption. Hence, we do not understand why the reviewer puts forward this argument to question our interpretations.

*A recent study by He et al., (2021, EPSL, mentioned by the authors) presented alkenone derived SST gradients from regions close to what the authors used to reconstruct their planktonic foraminiferal d18O gradients. However, the temporal development of these alkenone SST gradients is different from what the authors show from their d18O gradients.*

This is not true. The results of He et al. (2021) corroborate our stable-isotope-based paleoclimate interpretations. The black line in the plot below (i.e. Fig. 3C in He et al., 2021) shows the $TEX_{86}$ gradient between Sites U1461 (also in the northern Carnarvon Basin, just as Sites U1463 and 763) and U1459 (Perth Basin) in $TEX_{86}$ units. This low-resolution reconstruction of Leeuwin Current intensity by He et al. (2021) is in agreement with our high-resolution $\delta^{18}O$-based reconstruction (orange line). As the $TEX_{86}$ gradient is independent of temperature-calibrations, foraminiferal calcite diagenesis, or local $\delta^{18}O_{sw}$ changes, this co-variation provides strong support for our paleoclimatic interpretations, especially for the inferred period of weak Leeuwin Current between 3.7 – 3.1 Ma. This additional line of argumentation will be implemented in the revised version of the manuscript.

[Figure]

Comparison of the TEX$_{86}$ gradient between Sites U1461 and U1459 (Fig. 3C in He et al., 2021) and the isotopic gradient from our work. Both proxies are completely independent of each other, yet show similar patterns throughout the Plio-Pleistocene, indicating the robustness of these results.

**3. Detailed comments**

- Lines 21, 39, 280-288 will be rephrased
- Line 81-86: The point made in the study of He et al. (2011) is relevant because it challenges the current paradigm that Leeuwin Current is weaker during glacials compared to interglacials.
- Line 245-255 and first paragraph of §3.2 will be moved to the methods section, as suggested by the reviewer.
- Line 297-301: A similar comment was made by Clara Bolton (R1). More information on the assumptions on $\delta^{18}$Osw will be provided
- Line 334: Quantitative information will be provided
- Line 372 – 388: A similar comment was made by Clara Bolton (R1). It is true that our inferred habitat depth is rather deep, in closer agreement with the results of Rippert et al. (2016, equatorial Pacific) than with the results of Meinicke et al. (2021, West Pacific Warm Pool). The results of Meinicke et al. (2021) will be more explicitly discussed in §4.1, in conjunction with the discussion of the Rippert et al. (2016) results.
- Line 460-461 and 472-473: Results from cross-spectral analysis will be incorporated in the revised manuscript to convince the reviewer of time-series coherence on multiple timescales.
- Line 464-471: These indicators of relatively cold conditions in the Southern Hemisphere between 3.7 – 3.1 Ma will be added to Figure 8.
- Fig 1C: A similar comment was made by Clara Bolton (R1). This has to do with the resolution of the Copernicus Marine Service Information model, which seems to somewhat overestimate the water depth. This is no surprise, as Site U1459 is on a steep continental slope, and just a few kilometers further offshore, one encounters water depths >500 m. We will use a datapoint from the Copernicus Marine Service Information model higher-up on the slope, so that water depth is not exceeding 200 meters.

- Fig 3C: The distinction between new and previously-published data is clearly described in the text. We think it would make the figure unnecessary busy to indicate this here with different colors.
- Fig 7A: The long-term discrepancy is exactly our argument for delineating a secular period of weak Leeuwin Current between 3.7 – 3.1 Ma, coinciding with generally cooler conditions in the Southern Hemisphere.
- Fig. 8 will be redrawn with more distinct colors.

Cisneros-Lazaro, D., et al. (2022). Fast and pervasive diagenetic isotope exchange in foraminifera tests is species-dependent. *Nature Communications, 13*(1), 113. 10.1038/s41467-021-27782-8

De Vleeschouwer, D., et al. (2018). The amplifying effect of Indonesian Throughflow heat transport on Late Pliocene Southern Hemisphere climate cooling. *Earth and Planetary Science Letters, 500*, 15-27. https://doi.org/10.1016/j.epsl.2018.07.035

Meinicke, N., et al. (2021). Coupled Mg/Ca and Clumped Isotope Measurements Indicate Lack of Substantial Mixed Layer Cooling in the Western Pacific Warm Pool During the Last ~5 Million Years. *Paleoceanography and Paleoclimatology, 36*(8), e2020PA004115. https://doi.org/10.1029/2020PA004115

Rippert, N., et al. (2016). Constraining foraminiferal calcification depths in the western Pacific warm pool. *Marine Micropaleontology, 128*, 14-27. https://doi.org/10.1016/j.marmicro.2016.08.004

Stainbank, S., et al. (2020). Assessing the impact of diagenesis on foraminiferal geochemistry from a low latitude, shallow-water drift deposit. *Earth and Planetary Science Letters, 545*, 116390. https://doi.org/10.1016/j.epsl.2020.116390

---

## Author Comment (AC3)

**Final Author Comment**

De Vleeschouwer et al. – cp-2021-151

We found the review comment by Clara Bolton (R1) very useful in several aspects. First, it helps refining the presentation of one of the main points of the manuscript, (i.e., Leeuwin Current variability on secular timescales). Second, it prompts us to clarify ambiguities regarding $\delta^{18}O$-based temperature reconstructions. Third, it corrects a handful of editorial glitches. We warmly thank Clara Bolton for her constructive review of our work.

**1. Leeuwin Current intensity on secular timescales**

*I find it interesting that there is no clear unidirectional trend in isotopic gradient over the intensification of northern hemisphere glaciation interval, and LC current intensity on secular timescales thus seems to be almost independent of this (although LC intensity is apparently strongly linked to sea level changes at the G-IG scale) – this aspect is not discussed much.*

The reviewer refers to one of the key-messages of the paper: On orbital timescales, a significant portion of Leeuwin Current (LC) variability can be explained by sea level variations and orbital forcing of wind patterns. Yet, on secular timescales, LC variability seems partly decoupled from the global eustatic sea-level evolution. Our results do show a steepening of the isotopic gradient (hence a Leeuwin Current weakening) across the Plio-Pleistocene boundary. Yet, these steep isotopic gradients are not unique, as we report equally-steep isotopic gradients between 3.7 – 3.1 Ma.

We will implement two things to refine the presentation of this point, which is one of our main messages.

  a. First, we will add a paragraph in §4-2., in which we compare the two intervals of steep isotopic gradient (weak Leeuwin Current). The Pliocene interval between 3.7 – 3.1 Ma is characterized by a steep gradient with relatively low amplitude changes. The Pleistocene interval between 2.6 – 2.2 Ma is characterized by an equally-steep gradient but with higher amplitude variability.
  b. Second, we will highlight the fact that the SST records in the Agulhas/Benguela region exhibit similar behavior on secular timescales by means of cross-spectral analysis (see also answer to Referee #3). At those sites, early Pleistocene sea surface temperatures are similar or even somewhat warmer than late Pliocene (3.7 – 3.1 Ma) sea surface temperatures.

**2. Comments on temperature reconstructions**

*Figure 4a: perhaps add modern SST and T at proposed calcification depth of T. sacculifer (base of mixed layer) onto the Figure for comparison.*

We will follow this suggestion.

*Fig. 5: Calcification temperature reconstruction based on d18Oplanktic and Rohling Sea Level curve: I had a hard time assessing the robustness of this. Is the calculation of T in this way therefore based on the assumption that local d18Osw (i.e. the part of d18Osw related to local hydrological cycle effects and not global sea level) was constant? Is this assumption reasonable both on the long-term and on G-IG timescales (in the context of salinity/water mass changes associated with L current and upwelling changes in your study area) at your site? I think some discussion of expected local d18Osw changes on GIG timescales, and how this would impact the temperature reconstruction, would be nice.*

The calcification temperature calculation is based on a *global* $\delta^{18}$Osw reconstruction, assuming that *local* $\delta^{18}$Osw was constant. In the revised version of the manuscript, we will explicitly acknowledge and argue for this assumption. Based on the modern-day oceanography of the Perth Basin, we do not expect more than 0.6 psu salinity difference between phases of strong Leeuwin Current (Interglacial, present-day analogue: austral winter) and weak Leeuwin Current (Glacial, present-day analogue: austral summer). This estimate of salinity change on G-IG timescales is based on the present-day seasonal variability of the Leeuwin Current system. This analogy can be made because the Leeuwin Current is observed to be very active during austral winter, but very weak during summer (due to opposing wind patterns). Thus, the dynamic of the Leeuwin Current during glacials was likely quite similar to that of present-day austral summer, and vice versa. This analogy, however, likely represents an over-estimation of salinity change because the Leeuwin Current never shut down completely during glacial intervals.

If we were to include a salinity-driven local $\delta^{18}$Osw component, it would result in an increase of the amplitude of the $\delta^{18}$Osw reconstruction by about 0.10 – 0.15‰. This, in turn, leads to a reduction of the amplitude of the temperature reconstruction. The amplitude of possible salinity-driven $\delta^{18}$Osw variability is however small compared to the ice-volume-related change in $\delta^{18}$Osw. Moreover, salinity-driven variability is virtually unconstrained for the studied region and period, and the expected 0.6 psu amplitude is too small to be accurately reconstructed with SSS proxies. For all those reasons, we will continue calculating calcification temperatures solely based on a *global* $\delta^{18}$Osw reconstruction, assuming that *local* $\delta^{18}$Osw was constant. But, as mentioned before, we will acknowledge the underlying assumptions regarding local $\delta^{18}$Osw in the revised manuscript.

*In Figures 5 and 6, different sea-level reconstructions appear to be shown or used in calculations – Rohling 2014 in Fig. 5 and Rohling 2021 in Fig. 6. Is the more up-to-date sea-level record not also suitable for calculating the glacial contribution to d18Osw in Fig. 5?*

Figure 5 will be updated to use the Rohling et al. (2021) reconstruction for $\delta^{18}$Osw.

*It would be interesting to consider the G-IG amplitude of the G. sacculifer d18O signal in the context of contemporaneous and/or regional records from surface-dwelling species such as G. ruber, to see if this can give you any support for the proposed deep mixed layer depth habitat or relatively constant d18Osw conditions. Also, in the methods, it is not mentioned whether individuals with gametogenic calcite final chambers were avoided or included during picking.*

Contemporaneous records in the region have reported similar amplitude on G-IG timescales between *G. ruber* and *G. sacculifer*. However, Shackleton and Hall (1990), Karas et al. (2009) and Karas et al. (2011) all added a 0.25‰ inter-species offset for *G. ruber* to be comparable with *G. sacculifer*. This offset is in agreement with the interpretation that *G. ruber* is a surface dweller, while *G. sacculifer* thrives in the (bottom of the) mixed layer.

During picking, specimens with gametogenic calcite final chambers were avoided. This information will be added to the manuscript in §2.3.

*It may also be relevant to consider and compare the new paper by Meinicke et al. (2021, already cited) that also contains D47 measurements on Trilobatus trilobus and a deeper dwelling species for the Plio-Pleistocene from the Western Pacific Warm Pool - I think their interpretations re: depth habitat for T. trilobus are not the same as in this paper, are the two interpretations compatible?*

The two interpretations regarding the depth habitat of *Trilobatus trilobus*, formerly known as *G. trilobus* or *G. sacculifer* without sac-like final chamber are not exactly the same, but compatible.
    a.  Meinicke et al. (2021) determined the average calcification depth for *T. trilobus* to be roughly around 75 m (see their Fig 3a). Their data thus indicate an apparent calcification depth of *T. trilobus* at Site U1488 (West Pacific Warm Pool) within the lower mixed layer.
    b.  Rippert et al. (2016) report calcification depths around 120 m in the equatorial Pacific Ocean. This represents bottom of the mixed layer or top of the thermocline.
    c.  In our manuscript, we report clumped-isotope calcification temperatures for *T. trilobus* between 18-20°C. Compared to modern-day water column profiles at Site U1459, this corresponds to water depths between 100 – 150 m. Hence, our results are in closer agreement with the results of Rippert et al. (2016) than with the results of Meinicke et al. (2021). At Site U1459, *T. trilobus* calcification temperatures seem to indicate bottom of the mixed layer.
The results of Meinicke et al. (2021) will be more explicitly discussed in §4.1, in conjunction with the discussion of the Rippert et al. (2016) results.

*Line 349: This is interesting, I wonder is there any evidence in support of the hypothesis that Thaumarchaeota thrive/are more successful when phytoplankton biomass is low under oligotrophic conditions?*

We will back-up this statement by citing the work of Guo et al. (2021) (Frontiers in Marine Science, https://doi.org/10.3389/fmars.2021.715708).

*Line 361: Perhaps add that diagenesis (seafloor recrystallisation?) of foraminifera would specifically lead to a cool bias on D47 temperatures.*

The possibility of a cool bias on D47 temperatures has also been raised by Referee #2. A corresponding paragraph will be added to §4.1.

*Line 362: please briefly define "isotopic scrambling" for non-clumped specialists*

OK. We will mention that this implies the reset of the clumping of $^{13}C$ and $^{18}O$ atoms in $CaCO_3$.

*Line 498: Not clear how exactly low-resolution clumped isotope and TEX86 temperatures "demonstrate that the LC continued to operate […]" – do you deduce this from absolute temperatures? Or latitudinal gradients?*

From absolute temperatures. Without the Leeuwin Current, Site U1459 would be under the influence of the West Australian Current and SSTs would be <20°C. This reasoning will be added to the manuscript.

**3. Minor editorial comments**

- The last sentence of the abstract will be rephrased to emphasize that ITF forcing exerted a long-range influence on Southern Hemisphere climate throughout the Plio-Pleistocene.
- Carnarvon and Perth Basin will be labeled on the map in Fig. 2C
- The Figure 1 caption is indeed wrong: January and July will be flipped.
- The U1459 time-depth plot in Fig. 1C is for the locality of Site U1459. However, the spatial resolution of the bathymetry that underpins the Copernicus Marine Service Information model somewhat overestimates the water depth. This is no surprise, as Site U1459 is on a steep continental slope, and just a few kilometers further offshore, one encounters water depths >500 m. We will use a datapoint from the Copernicus Marine Service Information model higher-up on the slope, so that water depth is not exceeding 200 meters.
- We will specify equal weights for the two forcings in the composite.
- We will lay out more clearly that sub-surface upwelling of SAMW onto the shelf is the most likely process to explain the observed early-Pleistocene increase in paleo-productivity and organic carbon burial. This assessment is based on our observation that (1) $TEX_{86}$ SSTs are stable throughout the early Pleistocene, (2) we see an important cooling in D47-derived *T. trilobus* calcification temperatures (upper thermocline / bottom mixed layer), and (3) we see a simultaneous increase in U/Th (especially during glacials). This proxy co-evolution is most compatible with sub-surface upwelling of nutrient-rich waters when LC is weak. If productivity would be eddy-mixing driven, this would result in stable mixed layer temperatures because the overall LC would still be deep and stable over the site. But that is not what we observe, as we report an important drop in calcification temperatures of *T. trilobus* (both in $\delta^{18}O$ and D47).

Guo, J., et al. (2021). Variation of Isoprenoid GDGTs in the Stratified Marine Water Column: Implications for GDGT-Based TEX86 Paleothermometry. *Frontiers in Marine Science, 8*. 10.3389/fmars.2021.715708

Karas, C., et al. (2009). Mid-Pliocene climate change amplified by a switch in Indonesian subsurface throughflow. *Nature Geosci, 2*(6), 434-438.

Karas, C., et al. (2011). Pliocene Indonesian Throughflow and Leeuwin Current dynamics: Implications for Indian Ocean polar heat flux. *Paleoceanography, 26*. doi:10.1029/2010pa001949

Meinicke, N., et al. (2021). Coupled Mg/Ca and Clumped Isotope Measurements Indicate Lack of Substantial Mixed Layer Cooling in the Western Pacific Warm Pool During the Last ~5 Million Years. *Paleoceanography and Paleoclimatology, 36*(8), e2020PA004115. https://doi.org/10.1029/2020PA004115

Rippert, N., et al. (2016). Constraining foraminiferal calcification depths in the western Pacific warm pool. *Marine Micropaleontology, 128*, 14-27. https://doi.org/10.1016/j.marmicro.2016.08.004

Rohling, E. J., et al. (2021). Sea level and deep-sea temperature reconstructions suggest quasi-stable states and critical transitions over the past 40 million years. *Science Advances, 7*(26), eabf5326. doi:10.1126/sciadv.abf5326

Shackleton, N. J., & Hall, M. A. (1990). Pliocene oxygen isotope stratigraphy of Hole 709C. In R. Duncan, J. Backman, & L. Peterson (Eds.), *Proceedings of the Ocean Drilling Program, Scientific Results* (Vol. 115). College Station, TX: Ocean Drilling Program.

---

## Author Response (AR1)

**Author's response**

De Vleeschouwer et al. – cp-2021-151

Dear Prof. McClymont,

First of all, we want to thank you for granting us the necessary time to include some additional repeated measurements to answer some concerns raised by reviewer #3 on clumped isotopes statistics. Second, we thank all three reviewers for their assessment of our paper. We adapted the manuscript as described in the detailed replies to the three reviewers' comments (uploaded on February 15th 2022).

In this document, we include an overview of all relevant changes made to the manuscript in response to the reviewers' concerns.

**1. Leeuwin Current intensity on secular timescales**

Two important changes were made to the manuscript in order to improve the presentation of one of our main take-home messages:

a. First, we made changes to the text of paragraph §4.2. to emphasize the observation that the 3.7 – 3.1 Ma interval is characterized by isotopic gradients that are equally steep as the early Pleistocene interval between 2.6 – 2.2 Ma. This is important, as it illustrates that the 3.7 – 3.1 Ma interval is marked by a "weaker-than-expected" Leeuwin Current.

b. Second, we highlight the fact that the SST records in the Agulhas/Benguela region exhibit similar behavior on secular timescales: These records also reflect late Pliocene cooling between 3.7 – 3.1 Ma. This is now emphasized by means of grey rectangles on Figures 7 and 8. Contrary to what was written in the answer to the reviewers, we did not adopt cross-spectral analysis for this purpose because the records under investigation were of too low temporal resolution. But we believe that our visual approach in Figure 7 and 8 gets the message across.

These two changes help in bringing across a key message: The latitudinal isotopic gradient is a proxy for Leeuwin Current strength, and we observe a *stronger-than-expected* gradient (*weaker-than-expected* Leeuwin Current) in the late Pliocene. This time interval is then interpreted in terms of changing oceanography and southern hemisphere climate.

**2. Comments on temperature reconstructions**

Changes to the manuscript (especially in §3.2) were carried out as described in the answer to Reviewer 1.

**3. The impact of diagenesis on stable isotope results**

Changes to the manuscript (especially §4.1) were carried out as described in the answer to Reviewer 2. Section §4.1 now contains an explicit discussion of the potential effects of dolomitization and recrystallisation, and it provides an overview of all the actions and checks we adopted to minimize their effects on the clumped isotope and stable isotope results.

**4. Comments on the reliability of isotopic gradients for the reconstruction of Leeuwin Current strength**

Changes to the manuscript (especially §4.2) were carried out as described in the answer to Reviewer 2. A new supplementary figure (Fig. D1) is also provided in response to the reviewer's comment. This new figure contains a comparison of the $TEX_{86}$ gradient between Sites U1461 and U1459 (as in Figure 3C in He et al, 2021), and the isotopic gradient from this study. Importantly: these two gradients are completely independent of each other, as they were calculated from completely different proxies. Yet, they show similar patterns throughout the Plio-Pleistocene, corroborating the assumption that $TEX_{86}$ and isotopic gradients along the Leeuwin Current pathway are temperature-driven, and thus can serve as a proxy for Leeuwin Current intensity.

**5. TEX86 vs. clumped temperatures**

The seasonal warm-bias on TEX86 is now more explicitly considered as an important part of the proxy-discrepancy in the manuscript.

**6. Isotopic gradients**

Changes to the manuscript (especially in §3.2 and §4.2) were carried out as described in the answer to Reviewer 3.

**7. Clumped isotopes statistics.**

Additional measurements were carried out and appropriate changes to the manuscript were carried out. All the Reviewer #3 suggestions with respect to clumped isotope statistics were followed.

**8. Recalculated clumped isotope calibration**

The new paper by Peral et al. is still not accepted by *Geochemica Cosmochemica Acta*. In case *Climate of the Past* accepts our manuscript earlier than *Geochemica Cosmochemica Acta* accepts Peral et al. (2022), we can cite a pre-print of the paper by Peral et al., hosted on EarthArXive: https://doi.org/10.31223/X5VK82. This pre-print has already been cited in the current submitted version of our paper.

Kind regards,

David De Vleeschouwer and co-authors.